# Improving Instruction-Following in Language Models through Activation Steering

**Alessandro Stolfo**[1][*]  **Vidhisha Balachandran**[2]  **Safoora Yousefi**[2]

**Eric Horvitz**[2]  **Besmira Nushi**[2]

[1]ETH Zürich  [2]Microsoft Research

## Abstract

The ability to follow instructions is crucial for numerous real-world applications of language models. In pursuit of deeper insights and more powerful capabilities, we derive instruction-specific vector representations from language models and use them to steer models accordingly. These vectors are computed as the difference in activations between inputs with and without instructions, enabling a modular approach to activation steering. We demonstrate how this method can enhance model adherence to constraints such as output format, length, and word inclusion, providing inference-time control over instruction following. Our experiments across four models demonstrate how we can use the activation vectors to guide models to follow constraints even without explicit instructions and to enhance performance when instructions are present. Additionally, we explore the compositionality of activation steering, successfully applying multiple instructions simultaneously. Finally, we demonstrate that steering vectors computed on instruction-tuned models can transfer to improve base models. Our findings demonstrate that activation steering offers a practical and scalable approach for fine-grained control in language generation. Our code and data are available at https://github.com/microsoft/llm-steer-instruct.

## 1 Introduction

Instruction-following capabilities of large language models (LLMs) have enhanced their practical applications for real-world usage. These advances are powered by instruction-tuning methods (Ouyang et al., 2022; Bai et al., 2022; Wei et al., 2022; Sanh et al., 2022; Chung et al., 2024), which align the model's responses with user objectives, addressing the gap between pre-training and end-user needs (Askell et al., 2021). Instruction tuning allows users to specify constraints on attributes like format, tone, or length, which direct the model's behavior and output (Zhou et al., 2023b; Zhang et al., 2023; Lou et al., 2024). Gaining a deeper understanding of how LLMs internally represent and follow these instructions is essential for developing more controllable and reliable models.[1]

In this paper, we use a mechanistic method to investigate how language models internally represent various instructions and use these representations to influence and control the model's behavior. Prior research has shown that vector representations can be computed for tasks learned in context (Hendel et al., 2023; Todd et al., 2024; Liu et al., 2024) and various stylistic and semantic input features (Zou et al., 2023; Azaria & Mitchell, 2023; Zheng et al., 2024; Templeton et al., 2024; Marks & Tegmark, 2024, *inter alia*). These representations can be used for activation steering (Subramani et al., 2022): directly intervening on the model's activations to guide the generation. This approach has been successfully applied to control text attributes such as honesty (Li et al., 2023; Qiu et al., 2024; Zou et al., 2023), sentiment (Tigges et al., 2024), output style and tone (Turner et al., 2023; Liu et al., 2024; Scalena et al., 2024; von Rütte et al., 2024), harmfulness (Arditi et al., 2024; Wang & Shu, 2024), and sycophancy (Panickssery et al., 2024; van der Weij et al., 2024).

However, user instructions in generative tasks can be more complex and involve multiple parameters that need to be attended to and satisfied during generation (Sun et al., 2023). For example, users may

---

[*]Work done while at Microsoft Research. Coresp. to stolfoa@ethz.ch and besmira.nushi@microsoft.com.

[1]We use *instruction* to refer to specific constraints that can be added during interactive sessions with LLMs in a modular way to modify or extend a *base* question or request. We further elaborate on this definition in §2.

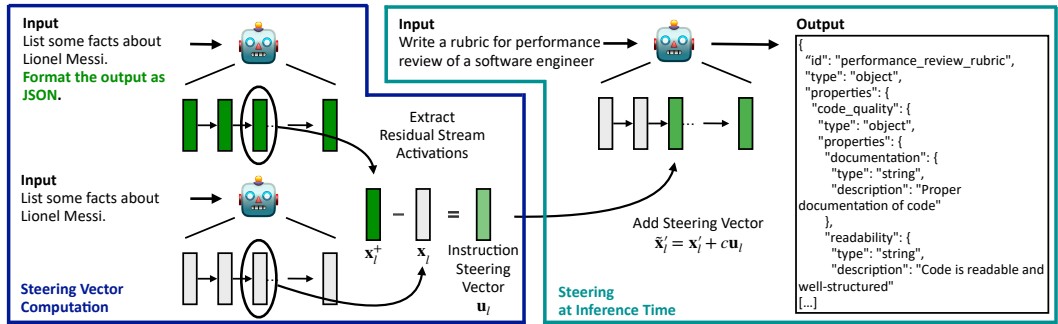

Figure 1: **Instruction Steering Process.** Steering vectors are computed as the difference in residual stream activations between inputs with and without the instruction. These vectors are then applied during inference to adjust the model's activations, guiding it to follow the desired instruction.

ask for a response to include bullet lists, have a certain number of sentences, or include/exclude specific content. The complexity of such instructions stems from the high number of possible variations, making it impractical to generate post-training data for all scenarios. In the quest for finding more efficient methods for controlling instruction following and guided by previous research in language model representations, we pose the question: Can we efficiently extract vector representations that encode and control specific instruction-following behavior?

We investigate this using a contrastive, additive steering method (Turner et al., 2023; Panickssery et al., 2024) that computes the difference in activations between inputs with and without an instruction (Figure 1, left), and applies this difference to guide the model to better follow instructions on new inputs (Figure 1, right). While previous work focused on high-level stylistic aspects, we target more diverse, verifiable instructions (Zhou et al., 2023a) that complement the base request's semantics (e.g., formatting, length constraints). These instructions are lower-level and more specific, with multiple possible instantiations (e.g., "Do not mention the word {keyword}"). It remains unclear whether models represent such instructions linearly and whether these representations can be used to reliably elicit specific behaviors, which would enable finer control of LLM outputs.

We conduct experiments using the Phi-3 (Abdin et al., 2024), Gemma 2 2B and 9B (Gemma Team, 2024), and Mistral 7B (Jiang et al., 2023) models, focusing on three types of instructions: output format (§3), output length (§4), and the inclusion/exclusion of specific words (§5). Our results on the IFEval dataset (Zhou et al., 2023a) provide evidence that vector representations can encode a wide range of instructions and enhance the model's instruction-following performance. Notably, we demonstrate that activation steering not only helps models follow constraints when no instruction is provided in the input, but it can also reinforce instruction adherence even when instructions are explicitly present, potentially countering *instruction drift* (Li et al., 2024). Furthermore, we show that it is possible to simultaneously steer for multiple constraints, such as controlling both format and length (§6). Finally, we present compelling evidence that cross-model steering–using vectors computed on an instruction-tuned model to steer a base model–is effective and can sometimes yield better adherence than same-model steering (§7), suggesting new possibilities for transferring task-specific skills across models, similar to "task arithmetic" (Ilharco et al., 2023).

Our work represents an important step toward operationalizing techniques from mechanistic interpretability to achieve practical improvements in scenarios with real-world utility.

## 2 STEERING FOR INSTRUCTION ADHERENCE

In this section, we define the types of instructions we consider (§2.1), introduce the methodology used to compute the vectors and the steering procedure (§2.2), and describe the experimental setup, including the data, metrics, and evaluation details (§2.3).

### 2.1 TYPES OF INSTRUCTIONS

The concept of instruction-following has been used to describe the broad capability of a model to answer any zero-shot query (Wei et al., 2022; Sanh et al., 2022). However, following prior work

(Zhou et al., 2023a;b; Sun et al., 2023), we adopt a more specific definition of instruction that refers to a constraint applied to a particular aspect of the model's output. These constraints are self-contained, modular, and can be imposed on various base queries. This definition decouples the ability of the model to follow the instruction from its factual knowledge and domain-specific skills. In our work, we focus on three specific types of instructions:

1. **Format instructions**, which dictate how the output should be presented. For instance, the model may be asked to produce responses in a specific format (e.g., "Provide the answer in JSON format") or highlight parts of the response in a particular way (e.g., "Use asterisks to emphasize at least two sections of the answer").
2. **Length instructions**, which specify the desired length of the output (e.g., "Answer using at most three sentences").
3. **Word-specific instructions**, which control the inclusion or exclusion of specific words or phrases in the output (e.g., "Do not include the word *AI* in the answer").

## 2.2 STEERING PROCEDURE

Activation (or representation) engineering involves constructing vectors of activation values which cause desired changes to output text when added to the forward passes of a frozen LLM (Zou et al., 2023). To identify a direction in the model's residual stream[2] that encodes information about a specific instruction, we use a technique called *difference-in-means* (Belrose, 2023). This method effectively isolates key feature directions in the model's internal activations (Marks & Tegmark, 2024; Tigges et al., 2024) and has been used to control various behaviors such as refusal and sycophancy (Arditi et al., 2024; Panickssery et al., 2024). We adapt this method to support finer-grained instructions with multiple possible instantiations (e.g., varying length, varying words to include and exclude). The process involves pairing two versions of the same request: one with only the base query (e.g., "List some facts about Lionel Messi") and another that additionally includes the instruction we want to represent (e.g., "List some facts about Lionel Messi, ensuring the output is valid JSON").

Let us denote these two inputs by $x$ (base query) and $x^+$ (base query with instruction), and consider a set of $N$ such pairs $(x_i, x_i^+)$, $i \in \{1, \ldots, N\}$. Let $\mathbf{x}_{i,l}, \mathbf{x}_{i,l}^+ \in \mathbb{R}^{d_{\text{model}}}$ be the values of the residual stream vector on the two queries at the last token of the input at layer $l \in \{1, \ldots, L\}$. We isolate the internal representation corresponding to the instruction by computing the difference in the residual stream vectors between the paired inputs. More formally, we compute a vector $\mathbf{u}_l \in \mathbb{R}^{d_{\text{model}}}$ representing the steering direction at layer $l$ for a given instruction as:

$$\mathbf{u}_l = \frac{\mathbf{v}_l}{\|\mathbf{v}_l\|}, \text{ where } \mathbf{v}_l = \frac{1}{N} \sum_i^N \left( \mathbf{x}_{i,l}^+ - \mathbf{x}_{i,l} \right). \tag{1}$$

Averaging over different base queries allows us to capture the activation values most closely associated with the instruction, independent of the base query. The computation of the steering direction is carried out using the representations at the last token of the input, which effectively encapsulate the model's behavior not only for the next-token-prediction task but also for the entire generation that follows (Todd et al., 2024; Scalena et al., 2024).

After identifying the steering direction, we compute the steering vector by re-scaling the unit vector $\mathbf{u}_l$ by a coefficient, $c \in \mathbb{R}$. For format instructions, we use a systematic scaling approach where the value of $c$ is selected to ensure that the residual stream activations are mapped to their mean value on inputs that contain the instruction in question. In particular, during a forward pass on a new example with residual stream values $\mathbf{x}' \in \mathbb{R}^{L \times d_{\text{model}}}$ at a given token position, we compute:

$$c = \bar{z} - \mathbf{x}_l'^{\text{T}} \mathbf{u}_l, \text{ where } \bar{z} = \frac{1}{N} \sum_i^N \mathbf{x}_{i,l}^{+\text{T}} \mathbf{u}_l. \tag{2}$$

This dynamic adjustment allows the model to effectively incorporate the constraint without over- or under-correcting its behavior. For length instructions, which have a more continuous nature, we experiment and show results with different values of $c$, illustrating their impact on the model's

---

[2]The model's residual stream is the per-token hidden state of dimensionality $d_{\text{model}}$ consisting of the sum of all previous component outputs (Elhage et al., 2021).

output. Finally, for word-specific constraints, we compute the weight using Eq. (2) and additionally perform a small grid search over neighboring values on a held-out set of examples to fine-tune the steering effect. The steering vector $c\mathbf{u}_l$ is then added to the corresponding residual stream layer and the forward pass is resumed with the updated residual stream value $\tilde{\mathbf{x}}'_l = \mathbf{x}'_l + c\mathbf{u}_l$. This procedure is carried out at a single layer across all token positions, motivated by previous findings that show models tend to deviate from instructions as they generate more tokens (Li et al., 2024). To find the optimal layer for steering, we perform a sweep across a subset of the model's layers, measuring the effect on a held-out set of queries and performing a perplexity-based quality check. Details about this procedure are provided in Appendix E.

## 2.3 EXPERIMENTAL SETUP

**Data.** We use an augmented version of the IFEval dataset (Zhou et al., 2023a), which consists of 25 distinct instructions, each paired with multiple base queries and expressed in different phrasings, for a total of 541 prompts. To evaluate format instructions, we focus on a subset of 163 examples that specifically relate to the output format. This subset includes instructions such as "The entire output should be wrapped in JSON format," requests to use a particular language (e.g., "Please respond using only the Punjabi language"), formatting requirements like "Wrap your entire response with double quotation marks," and casing instructions (e.g., "Answer using lowercase letters only"). A complete list of the instructions used is provided in Appendix B.[3] For length instructions, we generate prompts by concatenating base queries from IFEval to instructions derived from templates such as "Answer using at most $\{n\}$ sentences." For word-specific instructions, we use a subset of IFEval containing instructions about the inclusion or exclusion of keywords. The subset has 203 examples, and each example contains a prompt with a single keyword-related instruction. For steering vector computation and layer selection, we construct a separate set of synthetically generated prompts by combining base queries from IFEval with corresponding instruction descriptions to avoid test information leakage. Additional details about the data used are provided in Apps. C and D.

**Metrics.** To quantify the models' adherence to format instructions, we compute the "loose" instruction-following accuracy using the official IFEval evaluation script. For length instructions, we count the number of words or sentences in the model's output. For word-specific constraints, we verify the presence or absence of the specified keywords in the model's response, again using the IFEval evaluation script. We assess the statistical significance of differences in average instruction-following accuracy with and without steering using McNemar's test (McNemar, 1947). Results where steering leads to a significant improvement (p-value $< 0.01$) are marked with an asterisk (*).

In addition to assessing the model's instruction-following capabilities, it is important to verify that the model still effectively addresses the base query, even when generating under constraints. To measure the overall quality of the response, we set up a GPT-4o-based evaluation. For each base query $x$, GPT-4o generates a set of five yes/no questions aimed at assessing the quality of the response to $x$. For instance, given the query "Write an essay about the history of Martin Van Buren's presidency," GPT-4o generates questions such as "Does the essay provide context on the political, social, and economic climate during Van Buren's presidency?" and "Is the essay written in clear, grammatically correct English, and does it follow a logical structure?". Next, given the original query $x$, a model's response to $x$, and the generated questions we prompt GPT-4o to answer each question, provide a rationale for each yes/no response, or reply with "N/A" if the question is not applicable. We compute the proportion of questions answered positively for a given query and average this score across all queries to obtain a *response quality score*. We repeat this experiment 3 times and report the mean and standard error. Additional details are provided in Appendix F.[4]

**Evaluation.** We conduct experiments using the instruction-tuned versions of Phi-3 Mini (Abdin et al., 2024), Gemma 2 2B, 9B (Gemma Team, 2024), and Mistral 7B v0.1(Jiang et al., 2023). In addition, we investigate the transferability of steering vectors from the instruction-tuned to the base versions of Gemma 2 2B and 9B. All models are evaluated in a zero-shot setting, with outputs

---

[3]Although they are not strictly related to format, we include language constraints (e.g., "the answer should be in German") in this group, as, like other formatting instructions, they modify the surface-level presentation of the output without affecting the underlying content.

[4]Note that while the model's response may be influenced by specific instructions, these instructions are not provided to GPT-4o during the evaluation. The goal here is to focus the quality assessment solely on the comprehensiveness and relevance of the response with respect to the base query.

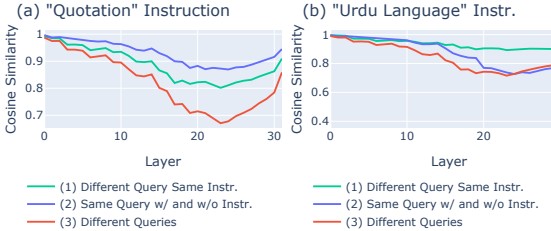

Figure 2: **Residual stream similarity across layers.** Phi-3's residual stream activations show higher cosine similarity between examples with the same instruction compared to those without, indicating effective capture of instruction-relevant features.

Table 1: **Steering Vector Projections.** The projection onto the vocabulary space of the contrastively-computed vectors promotes tokens that are semantically related to the respective instruction.

| Instruction | Layer | Top Tokens |
|---|---|---|
| JSON Format | 18 | _[{, _json, _`{, _JSON |
| Capitalize | 28 | _PRO, _TH, _ FOR, _AND |
| Highlight Text | 26 | *`, _*, _*, (*, *\\ |
| Lowercase | 18 | _lower, _lowest, _russ |
| Bullet List | 28 | *), _*, *, _*), •, */ |
| Quotation | 26 | _`", _'", _"", _"; _"</ |
| Urdu Language | 24 | _Islam, _Pakistan, _Pak |
| Hindi Language | 18 | _Indian, raj, _India |
| German Lang. | 16 | _die, _im, _dies, _gener |

decoded greedily. We assess our method's performance by steering the model under two input settings: (1) with text instructions provided in the input, and (2) without any text instructions. These settings allow us to explore two questions: Are the computed steering vectors informative enough to guide the model's behavior even without explicit instructions? And can steering further improve performance when instructions are provided in the input?

## 3 FORMAT INSTRUCTIONS

**Representational Analysis.** To assess whether the model effectively captures instruction-related information in its internal representations, we analyze Phi-3's residual stream activations at the last token of the input by calculating cosine similarity across three sets of inputs: (1) pairs of inputs that share the same base query, one with and one without the instruction; (2) inputs with different base queries but the same instruction; and (3) inputs with different base queries and no instruction. High similarity in set (2) would suggest that the model captures a shared feature (the instruction), while we expect set (1) to show relatively high similarity due to shared base queries, and set (3) to show lower similarity due to the absence of any shared features. Figure 2 shows how these measures evolve across layers for two instructions. For the "quotation" instruction, where the model is asked to wrap the output in quotation marks, there is a clear difference between sets (2) and (3), indicating that the instruction is partially captured (green vs. red lines in Figure 2a). For the "Urdu language" instruction, set (2) shows higher similarity than set (1), suggesting strong representation of the instruction (green vs. blue lines in Figure 2b). These results indicate that the model can effectively encode instruction-relevant features at the last input token, which supports steering model behavior based on specific constraints. Additional representational analyses are provided in Appendix H.

**Steering Vector Computation.** We compute steering vectors for each of the 12 format-related instructions in the IFEval subset and for the 19 language-based instructions specified in the dataset. During the selection of the optimal steering later, we compare the validation score with and without steering to ensure that the steering intervention leads to an improvement. If no layer shows an improvement in the validation score, no steering is applied at test time. Details about the layers selected for steering are provided in Appendix E. To ensure that the steering vectors effectively capture the information related to the instructions, we inspect them by projecting the vectors onto the model's vocabulary space (nostalgebraist, 2020; Geva et al., 2022). We project the vectors using the model's unembedding matrix and examine the vocabulary tokens with the highest logit values. Table 1 presents the top tokens associated with several of the instructions we consider. We observe that the tokens promoted by the steering vectors are semantically related to the intended instruction, providing an initial validation that the vectors are capturing the desired features.

**Steering Results.** We first evaluate steering on inputs without explicit text instructions (Figure 3a). In this setting, the instruction-following accuracy without steering hovers around 10% as the input has no information about the instruction. This non-zero accuracy reflects cases where the models incidentally satisfy the instruction, such as when a model rephrases a sentence without using commas, thus accidentally meeting the "no comma" constraint. When steering is applied, we observe a consistent increase in instruction adherence across all models, with accuracy improving to approximately 30%. This shows that the steering vectors encode meaningful information of the instructions

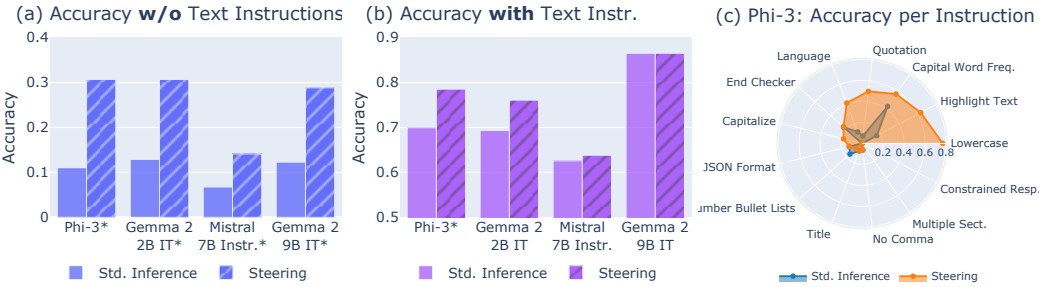

Figure 3: **Format Instructions.** (a) Instruction-following accuracy without explicit text instructions shows significant improvement with steering across all models. (b) Steering enhances accuracy even when text instructions are provided. (c) Per-instruction accuracy for Phi-3 without text instructions.

and can be effectively used to steer the models toward the intended behavior. Next, we evaluate on inputs with instructions provided in the text (Figure 3b). As expected, the instruction-following accuracy without steering is higher in this setting, ranging between 60% and 90%. Nevertheless, steering still results in a notceable performance boost for two out of four models, demonstrating that steering can enhance instruction adherence even when the instructions are explicitly given.

To further analyze which instructions benefit most from steering, we break down Phi-3's performance by instruction on input without explicit text instructions (Figure 3c). Notably, we observe significant improvements for instructions such as "Lowercase," which requires the output to be entirely in lowercase characters, and "Highlight Text," which asks the model to emphasize parts of the response using markdown syntax. However, certain instructions display variability in steering effectiveness. For instance, the "End Checker" instruction requires the model to finish the response with a variable, input-dependent sentence (e.g., "Hope you agree with me," or "Is there anything else I can help you with?"). Because the steering vector is computed by averaging over all examples containing this instruction, it likely fails to capture the specific sentence required in each individual case, reducing its ability to steer the model toward precisely following this type of instruction.

**Response Quality Evaluation.** Figure 4 compares the average differences in response quality scores for the four models. The two leftmost groups of bars show the score changes due to steering, with and without input instructions (calculated only for cases where steering is applied). Satisfying specific constraints during generation may lead to less comprehensive responses and hence following instructions is expected to impact the response quality. To show this effect, we also present the differences in quality scores observed when explicit text instructions are added to the base queries without steering. We observe small decreases in quality score due to steering in the setting including instructions (with the exception of Gemma 2B) while the decreases in the setting without instructions are slightly larger but comparable to the effect caused by simply adding the instructions as text. However, we also observe instances where steering compromised the quality of generation– examples where the model generated nonsensical tokens or repeated itself. These failures likely result from suboptimal steering or partially insufficient objectives during the search for layers to intervene, as steering for specific properties can

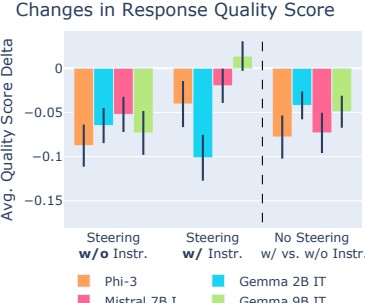

Figure 4: **Quality Score.** Average response quality score changes for four models, comparing the effects of steering with and without input instructions to the effect of adding instructions without steering. Steering decreases quality similarly to adding instructions.

increase perplexity or reduce fluency (Turner et al., 2023; Stickland et al., 2024). We provide some examples from the few cases we observed in Appendix F.

## 4 LENGTH INSTRUCTIONS

**Steering Vector Computation.** Length constraints can be specified in various ways, such as by the number of sentences, words, or lines. However, unlike format instructions, it seems impractical to compute a separate steering vector for each possible length constraint. Instead, we focus on captur-

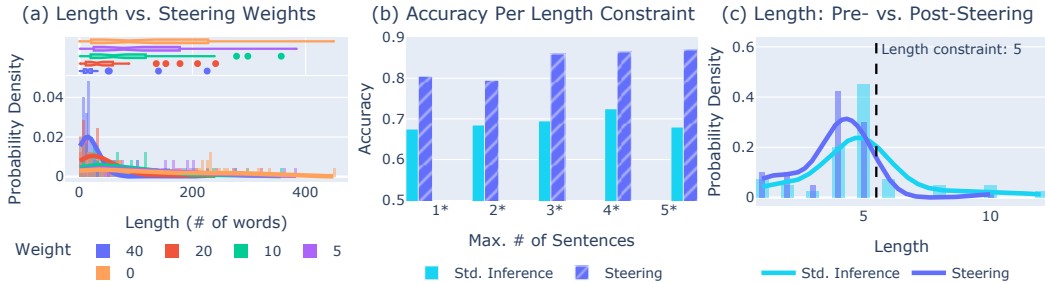

Figure 5: **Length Instructions.** (a) Modulating the steering weight $c$ effectively adjusts response length, with larger values leading to more concise outputs. (b) Steering enhances adherence to the maximum length constraints when they are explicitly specified in the input text. (c) Shift in the response length distribution upon steering for outputs constrained to a maximum of 5 sentences.

Table 2: **Example of Steering for Length Control.** Responses to the same prompt with increasing steering weights, resulting in progressively shorter outputs in terms of sentence and word count.

| Weight | Resp. to "Write a movie plot that involves dream, fist fighting, and superpower." | # of Sents. | # of Words |
|---|---|---|---|
| 0 | In the bustling city of New Haven, a young, introverted artist named Alex dreams of a world where his art comes to life. One night, Alex has a vivid dream where he discovers he possesses a unique superpower: the ability to bring [...] | 15 | 324 |
| 5 | In a world where dreams can be harnessed as a source of power, a young, introverted artist named Leo discovers he has the unique ability to enter and manipulate the dreams of others [...] | 9 | 228 |
| 10 | In a world where dreams can manifest into reality, a young woman named Elara discovers she possesses the rare ability to fight with her fists in her dreams [...] | 10 | 203 |
| 20 | In a world where dreams can manifest reality, a young girl named Lila discovers she can unleash her latent superpower through her dreams [...] | 3 | 81 |
| 40 | A young girl dreams of a superpower, fights a rival in a dream, and unleashes her power to save her village from destruction. | 1 | 23 |

ing more general notions of *conciseness* and *verbosity*. To achieve this, we compute vector representations for instructions that prompt the model to be brief (e.g., "The answer should be brief") or to provide more detailed responses (e.g., "Provide a long answer"). We synthetically generate these prompts by appending such instructions to base queries from IFEval. After computing these steering vectors, we evaluate the model on a separate set of IFEval base queries, this time appending length instructions specified in terms of the number of sentences (e.g., "Answer using at most 3 sentences").

**Steering Results.** Unlike format instructions, steering the model for length constraints allows for continuous modulation, enabling interpolation between varying degrees of conciseness or verbosity. To explore this, we manually adjust the steering weight $c$ and examine how this affects the response length. On a set of 50 base prompts without any explicit length instructions, we generate responses from Phi-3 using different values of $c$, measuring the distribution of output lengths. As shown in Figure 5a, increasing the value of $c$ effectively shortens the model's responses: larger steering weights produce increasingly concise outputs. We provide an example of different outputs generated on the same inputs with different steering weights in Table 2. Next, we assess whether steering for length is effective even when the model is provided explicit length instructions. Using a set of 200 base prompts, we introduce instructions that request the model to limit its response to a maximum of $n \in \{1, \ldots, 5\}$ sentences. First, we evaluate how often the model adheres to this constraint without steering (light blue bars in Figure 5b). Then, on the same inputs, we apply the steering vector for conciseness with a weight of $c = 20$ (dark blue bars in Figure 5b). Across all five values of $n$, we observe a significant and consistent improvement in how often the model's responses comply with the length constraint. Figure 5c further illustrates how steering shifts the response length distribution toward shorter outputs while still allowing for variability and avoiding overly short responses. We also conducted experiments with steering for longer outputs and for exact output length, observing positive results (Appendix J). Finally, response quality evaluation results are provided in Appendix F.

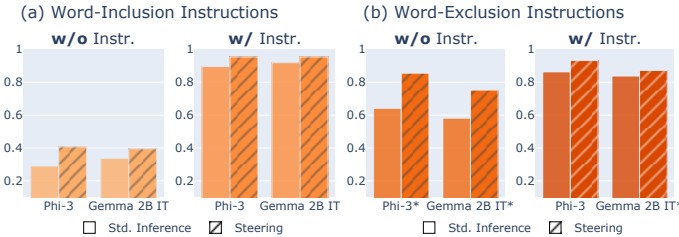

Figure 6: **Word-specific Instructions.** (a) Steering improves keyword inclusion accuracy for Phi-3 and Gemma 2 2B IT, with and without explicit instructions. (b) Negative steering reduces the occurrence of undesired keywords, in both settings.

Table 3: **Example of Word Exclusion.** The instruction alone fails to exclude "step," but steering successfully removes it.

| Steer | Resp. to "What are the steps to be followed for the documentation of a GM in SAP? Do not use the word *step*." |
|---|---|
| No | Documenting a General Master in SAP involves [provides guide] By following these **steps**, you can create comprehensive [...] |
| Yes | ## Documenting a General Master (GM) in SAP: [provides guide] By following this **comprehensive guide**, you can create a well-structured [...] |

# 5   WORD-SPECIFIC INSTRUCTIONS

**Word Inclusion.** For this set of instructions, we compute word-specific steering vectors. To generate these vectors, we append different phrasings of a request to include a specific word $w$ (e.g., "please include the word $\{w\}$ in your response") to a set of base queries. These prompts are used to compute a vector representation for the "include word $\{w\}$" instruction. While this requires a separate steering vector for each keyword at inference time, the vectors can be generated on-the-fly using arbitrary base queries unrelated to the keyword itself. Furthermore, these vectors can be computed with a small number of examples; in our experiments, we use only 20 examples. The models are then evaluated on the subset of IFEval that contains keyword inclusion/exclusion constraints.[5] The results, presented in Figure 6a, demonstrate the effectiveness of steering for word inclusion. By applying the word-specific steering vectors, we observe a notable increase in the frequency with which the model successfully includes the requested keywords in its responses.

**Word Exclusion.** For keyword exclusion, we initially used the same procedure, appending instructions like "ensure the word $w$ does not appear in the answer" to base queries and computing the corresponding steering vectors. However, upon inspecting the steering vectors, we found that projecting these vectors onto the vocabulary space resulted in high logit values for the tokens corresponding to the words meant to be excluded. In other words, adding these vectors to the model's residual stream actually increased the probability of generating the very keywords we intended to exclude. To address this issue, instead of computing exclusion vectors, we compute the vectors for inclusion and then subtract them from the model's residual stream. This technique effectively steers the model away from using the words captured by the vector. Figure 6b shows the results of steering for keyword exclusion using this approach. We observe that subtracting the inclusion vectors significantly reduces the frequency of the undesired keywords in the model's responses, both in cases where no textual instructions are present and where explicit exclusion instructions are provided in the input. Table 3 reports an example where the instruction to exclude a specific word is ineffective on its own, but applying steering successfully enforces the constraint.

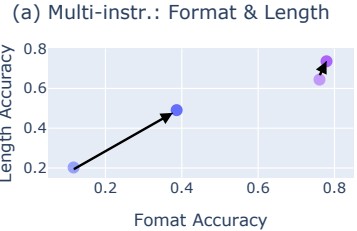

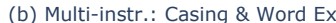

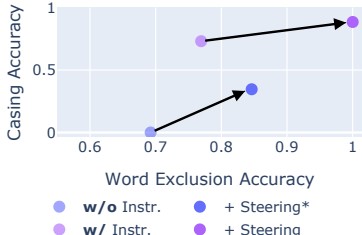

Figure 7: **Multi-instruction Steering.** Steering for two instructions improves adherence to both.

# 6   MULTI-INSTRUCTION STEERING

Next, we present results on using our steering approach to handle multiple instructions simultaneously. With the same method as in the previous experiments, we steer the Phi-3 model for two

---

[5]IFEval prompts may contain instructions for the inclusion or exclusion of multiple keywords in the same example. We separate such cases into inputs with single-keyword instructions.

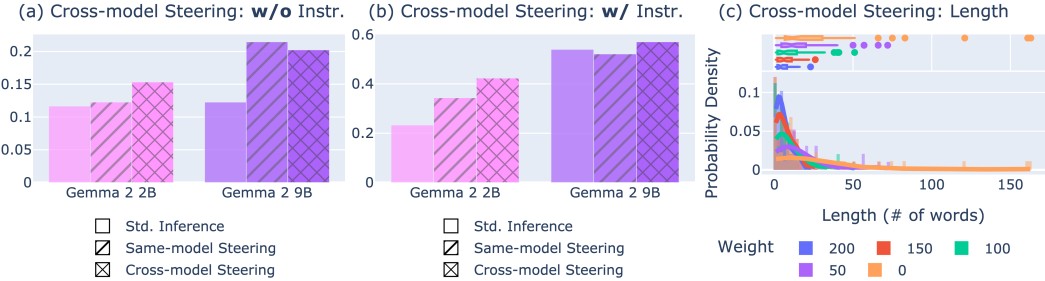

Figure 8: **Cross-model Steering.** (a,b) Steering vectors from instruction-tuned models transfer effectively to base models, with cross-model steering outperforming same-model steering in Gemma 2 2B. (c) Cross-model steering for length instructions shortens output as steering weight increases.

instructions at once, applying the steering vectors at the layers previously identified as optimal for each individual instruction. We opt for injecting multiple steering vectors simultaneously at different locations instead of combining multiple steering vectors into a single one, as previous work has shown the latter approach to be largely unsuccessful (van der Weij et al., 2024).

Figure 7 shows results from experiments with steering two instructions the same time: format (all 13 instructions) and length (Figure 7a), as well as lowercase and keyword exclusion (Figure 7b). In both cases, steering leads to improvements on both axes. These findings suggest that steering for multiple instructions at once is feasible and can lead to performance gains across different constraints. However, we anticipate that issues may arise in certain cases, particularly when dealing with conflicting instructions, where further refinement may be necessary to balance competing constraints.

## 7    CROSS-MODEL STEERING

Instruction-tuned models show improved instruction-following abilities, suggesting they may also form more effective instruction representations. This raises the question: can we leverage the representations computed on instruction-tuned models to steer a base model more effectively? This idea is motivated by prior work showing that fine-tuning reinforces existing mechanisms in language models (Prakash et al., 2024) and that instruction tuning causes minimal weight changes (Lee et al., 2024; Jain et al., 2024). Additionally, linear representations derived through mean activation difference and sparse autoencoders (Huben et al., 2024) have been shown to be transferable between base and chat models (Panickssery et al., 2024; Kissane et al., 2024). To study this in our setting, we conduct experiments with the base and instruction-tuned versions of Gemma 2 2B and 9B. We focus on format instructions, using the same data and procedures outlined in §3. The key difference is that we apply steering vectors computed on the instruction-tuned models to steer the base models, comparing this approach to same-model steering (vectors computed on the base model).

In both the base query-only setting (Figure 8a) and with explicit instructions (Figure 8b), steering base models using vectors computed on instruction-tuned counterparts yields performance gains that demonstrate the transferability of instruction representations. Notably, in the case of Gemma 2 2B, cross-model steering outperforms same-model steering across both settings, highlighting its potential for more effective instruction adherence in smaller models. We also explore cross-model steering for length instructions. As described in §4, we compute a steering vector for conciseness from the instruction-tuned version of Gemma 9B and apply it to the base version. Figure 8c shows that increasing the steering weight shortens the outputs, demonstrating that representations learned in instruction-tuned models can still meaningfully modulate the output in the base model's space.

This is the first demonstration that cross-model steering–using vectors from instruction-tuned models–can outperform same-model steering in base models. This finding suggests that specialized representations from fine-tuned models can be leveraged to steer base models more effectively, opening new possibilities for composable transfer learning in instruction-based tasks where task vectors may originate from different models that are instruction-tuned in specialized domains.

## 8 RELATED WORK

**Instruction Following.** Training models to follow instructions is crucial for improving LLM performance and ensuring safe deployment, with various methods developed to enhance instruction adherence (Ouyang et al., 2022; Sanh et al., 2022; Wei et al., 2022; Bai et al., 2022; Chung et al., 2024), and datasets designed to train and evaluate instruction-following behavior (Ye et al., 2021; Wang et al., 2022; Gupta et al., 2022; Finlayson et al., 2022; Mishra et al., 2022; Longpre et al., 2023; Köpf et al., 2023). Natural language instructions have demonstrated significant promise in providing fine-grained control over model outputs (Zhou et al., 2023b). However, capable models still struggle with tasks that require outputs to satisfy fine-grained, hard constraints (Sun et al., 2023) and tend to drift from adhering to a constraint as the generation lengthens (Li et al., 2024). Motivated by these challenges, our work investigates how to improve instruction-following behavior by directly intervening on the model's activations at inference time.

**Language Model Representations.** Our approach is inspired by prior research that shows it is possible to obtain vector representations encoding information about tasks on which a language model has been trained (Ilharco et al., 2023; Huang et al., 2024) or tasks learned in context (Hendel et al., 2023; Todd et al., 2024). These studies are part of a broader body of work that examines the linear representation of features such as truthfulness (Li et al., 2023; Azaria & Mitchell, 2023; Marks & Tegmark, 2024), sentiment (Tigges et al., 2024), harmlessness (Zou et al., 2023; Zheng et al., 2024), sychophancy (Perez et al., 2023; Panickssery et al., 2024; Sharma et al., 2024), factual knowledge (Gurnee & Tegmark, 2024), and refusal (Arditi et al., 2024). In addition, recent works have employed sparse autoencoders to identify feature directions in an unsupervised manner (Bricken et al., 2023; Huben et al., 2024; Templeton et al., 2024). A shared hypothesis across these works is that LLMs represent features or concepts as linear directions in activation space (Mikolov et al., 2013; Bolukbasi et al., 2016; Elhage et al., 2021; Nanda et al., 2023; Park et al., 2024; Olah, 2024). While recent studies suggest that not all features may be linearly encoded (Engels et al., 2025; Csordás et al., 2024), the linearity assumption has been effective for both concept erasure (Ravfogel et al., 2020; Belrose et al., 2023; Shao et al., 2023; Guerner et al., 2024) and model steering.

**Model Steering via Activation Editing.** It is well-established that the generation of a language model's output can be manipulated by directly editing activation values during inference (Dathathri et al., 2020; Subramani et al., 2022). Recent studies have shown that this approach can effectively steer models to be more honest and truthful (Li et al., 2023; Qiu et al., 2024), sycophantic (Panickssery et al., 2024; van der Weij et al., 2024), morally aligned with human values (Zou et al., 2023; Lu & Panickssery, 2024), or to display different sentiments, output styles, and languages (Turner et al., 2023; Liu et al., 2024; Tigges et al., 2024; Scalena et al., 2024; von Rütte et al., 2024). Steering methods have also been used to control the model's uncertainty (Rahn et al., 2024), adopt different personas (Cao et al., 2024), provide alternative factual answers (Hernandez et al., 2024), and respond to harmful requests (Arditi et al., 2024; Wang & Shu, 2024). Similarly to some of these works (Burns et al., 2023; Turner et al., 2023; Panickssery et al., 2024; Arditi et al., 2024; van der Weij et al., 2024), we compute steering vectors based on input pairs that differ by a specific feature—in our case, the presence or absence of instructions. However, while previous studies have focused on high-level concepts such as sentiment, style, and safety, we focus on lower-level, hard constraints defined through natural language instructions, allowing for finer-grained control of the model's output.

## 9 CONCLUSION

We demonstrated the effectiveness of activation steering for improving language models' adherence to instructions. By computing steering vectors based on activation differences between inputs with and without instructions, we guide models to follow constraints related to format, length, and word inclusion/exclusion. These vectors capture meaningful instruction representations, enabling models to meet constraints even without explicit instructions, while also enhancing performance when instructions are present. Additionally, we show that models can handle multiple constraints simultaneously, offering a flexible framework for controlled language generation. Finally, we explore cross-model steering, revealing that vectors computed on instruction-tuned models can improve the behavior of base models, in some cases surpassing same-model steering.

## REPRODUCIBILITY STATEMENT

We submitted our code and data as supplementary materials, and they will be open-sourced upon publication. §2.2 outlines our methodology in detail. Information about the datasets we used, as well as the procedures for obtaining and augmenting them, is provided in §2.3, along with Apps. C and D. §2.3 also contains the details of the evaluation metrics we employed and the process for generating model outputs. Appendix F provides further details on the implementation of our quality score metric. Additional implementation and experimental details are included in Appendix G.

## ACKNOWLEDGMENTS

We would like to express our gratitude to Andy Arditi, Ahmed Awadallah, Natasha Butt, Varun Chandrasekaran, Arthur Conmy, Nico Daheim, Saibo Geng, Suriya Gunasekar, Neel Joshi, Siddharth Joshi, Mazda Moayeri, Neel Nanda, Harsha Nori, Kyle O'Brien, Vibhav Vineet, and Vilém Zouhar for their valuable feedback and insightful discussions throughout the development of this project. We also extend our thanks to the AI Frontiers group at Microsoft Research, with special recognition to Ece Kamar for practical support. Alessandro acknowledges the support of armasuisse Science and Technology through a CYD Doctoral Fellowship.

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

## A  LIMITATIONS AND FUTURE WORK

While our work demonstrates the potential of activation steering for improving instruction adherence, it has multiple limitations that highlight avenues for future research. Activation steering involves several degrees of freedom, including the selection of the model layer to intervene at, determining the steering weight, and identifying where in the input or output sequence to apply the intervention. While our method systematically selects these parameters based on paired examples that differ by the presence of an instruction, it does not fully explore the space of possible interventions, leaving room for optimization. For instance, the quality of instruction representations directly impacts the effectiveness of steering. Representations computed from inputs where the model fails to adhere to instructions may reduce the efficacy of steering. Future work could explore filtering out such representations during vector computation to improve results, although this would require additional instruction-following accuracy checks, which our current approach avoids. Furthermore, our method applies a fixed steering weight throughout the generation process. Dynamically modulating the steering weight during generation (Scalena et al., 2024) could adapt the intervention to the evolving context and improve performance.

Another area for exploration is the effect of the number of examples used to compute steering vectors. Previous research indicates that clean concept representations can be obtained with relatively few examples (e.g., 128; Arditi et al., 2024). In our experiments, steering vectors computed from as few as 20 examples proved effective. However, we did not extensively investigate how varying the size of the example set affects the quality of these vectors. Future research could focus on optimizing this balance between computational efficiency and steering performance.

While we consider a reasonably large and diverse set of instructions in our experiments, the range of possible real-world user requests is vast and cannot be exhaustively represented. Additionally, our use of perplexity (computed with GPT-2; Radford et al., 2019) as a proxy for output quality focuses on detecting issues in fluency and coherence but may miss subtler forms of degradation, such as factual inaccuracies or logical inconsistencies. Incorporating more nuanced quality metrics or task-specific evaluations could improve parameter selection.

Finally, an interesting direction for future work stems from our cross-model steering experiments. In our work, we experiment with transferring steering vectors from instruction-tuned to base models of the same family and parameter size. An interesting avenue for future research is exploring the transferability of steering vectors across models with different architectures or sizes, potentially by learning transformations that map the latent space of one model to another.

## B  LIST OF INSTRUCTIONS

In Table 4, we provide a list of the instructions and examples used in the subset of the IFEval dataset for our experiments about the output format (§3). For the "language" instructions, the dataset includes the following languages: German, Urdu, Portuguese, Korean, Marathi, Punjabi, Kannada, Farsi, Swahili, Russian, Hindi, Arabic, Nepali, Telugu, and Gujarati.

## C  EXTRACTING IFEVAL BASE QUERIES

To obtain base queries without instructions, we use GPT-4o (OpenAI, 2024)[6] to strip the instructions from the base queries, as the IFEval dataset does not explicitly annotate the instructions within the prompts. We use a one-shot prompt a fixed in-context example. The prompt used is reported in Table 5. These base queries are then used to generate additional base query-instruction combinations, augmenting the original dataset. Further details on the data used for steering vector computation and evaluation are provided in Appendix D.

Table 4: **Format-Related Instructions and Examples.** Format-related instructions used in our experiments, along with examples illustrating the specific constraints the model is expected to follow.

| Instruction | Example |
|---|---|
| No Comma | You are not allowed to use any commas in your response. |
| Lowercase | Please ensure that your response is in English, and in all lowercase letters |
| Language | Please respond using only the Kannada language, no other language is allowed. |
| JSON Format | Wrap the entire output in JSON format. You can use markdown ticks such as "'. |
| Quotation | Wrap your entire response with double quotation marks. |
| Multiple Sections | Write a 4 section [base query]. Each section should be explicitly noted as Section X. |
| Number of Bullet Points | Your answer must contain exactly 6 bullet point in Markdown using the following format:\n* Bullet point one.\n* Bullet point two.\n... \n* Bullet point six |
| Highlighted Sections | At least 15 sections should be highlighted with markdown such as *highlighted section*. |
| Title | Your answer must have a title contained in double angular brackets, such as <<title>>. |
| Capitalize | Make sure your entire response is in English, and in all capital letters. |
| Capital Word Frequency | Use words in all capital letters at least 3 times to highlight key points. |
| End Checker | The very end of your response should read "You cannot fail with the steps listed above." No other words should follow this phrase. |
| Constrained Response | Answer with exactly one of the following phrases: "My answer is yes.", "My answer is no.", "My answer is maybe." |

Table 5: **Prompt for Instruction Removal.**

```
Given a question that imposes a set of constraints on the answer, make the question
simpler by removing all the constraints.  You will be given the original question
and a set of constraints to remove from it, and should output the simplified question
with the constraints removed.  Nothing else should be removed other than the listed
constraints.
For example:

<original_question>
I am planning a trip to Japan, and I would like thee to write an itinerary for my
journey in a Shakespearean style.  You are not allowed to use any commas in your
response.
<\original_question>

<constraints>
[punctuation:no_comma]
<\constraints>

<output>
I am planning a trip to Japan, and I would like thee to write an itinerary for my
journey in a Shakespearean style.
<\output>

<original_question>
Write a 300+ word summary of the wikipedia page https://en.wikipedia.org/
wiki/Raymond_III_Count_of_Tripoli.  Do not use any commas and highlight at least 3
sections that has titles in markdown format, for example *highlighted section part
1*, *highlighted section part 2*, *highlighted section part 3*.
<\original_question>

<constraints>
[punctuation:no_comma, detectable_format:number_highlighted_sections,
length_constraints:number_words]
<\constraints>

<output>
```

Table 6: **Prompt for Instruction Addition.**

```
Given a simple question, we want to make the question a bit harder by adding
constraints to the way it can be answered.  You will be given the original question
and a constraint to add to it, and should output the harder question with the
constraint integrated into it.  Nothing else should be added or removed from the
question.  Only the constraint should be added to the question.
For example:

<original_question>
The opposite of youth is not age, but ...?
<\original_question>

<constraints>
[detectable_format:number_highlighted_sections]
<\constraints>

<output>
The opposite of youth is not age, but ...?  Highlight at least 2 sections in your
answer with markdown, i.e.  *highlighted section*.
<\output>

<original_question>
Write a 300+ word summary of the wikipedia page https://en.wikipedia.org/
wiki/Raymond_III_Count_of_Tripoli.  Do not use any commas and highlight at least 3
sections that has titles in markdown format, for example *highlighted section part
1*, *highlighted section part 2*, *highlighted section part 3*.
<\original_question>

<constraint>
detectable_format:number_highlighted_sections {'num_highlights':  2}
<\constraint>

<output>
```

# D   DATA AND VECTOR COMPUTATION DETAILS

**Format Instructions.** The steering vectors and layer selection for format instructions are computed using a separate set of synthetically generated prompts. To generate a synthetic dataset containing all base prompts with all instructions from the IFEval dataset, we start with the base prompts as described in Appendix C. We then create an augmented version of the IFEval dataset by combining each base prompt with every available instruction. This is done by prompting GPT-4o using in-context examples, as shown in Table 6. When adding a constraint to a base prompt, we include a randomly selected in-context example of the same type of constraint from the single-constraint dataset. These single-constraint examples are generated by prompting GPT-4o to remove all but one instruction from each prompt. For this, we use a fixed, manually curated in-context example. This procedure is carried for each base query and each format instruction. We then filter out any base query and instruction combinations that appear in the evaluation set to ensure that the base queries used for steering vector computation and layer selection do not overlap with those in the evaluation prompts. This process yields approximately 450 examples for each instruction, which are used for both the steering vector computation and for validation. We provide examples of data generated using this procedure in Table 7.

For validation, we use a set of 96 examples (8 per instruction), sampled from the synthetically data described above. For format instruction evaluation, we utilize the format-related subset of IFEval queries (163 in total), which includes the 13 distinct instructions detailed in Appendix B.

**Length Instructions.** For length instructions, the steering vectors are computed using a fixed set of 50 IFEval base queries, obtained as described in Appendix C. To this set, we append various manually-annotated phrasings of instructions that request the model to generate concise responses, such as "Be concise" or "The answer should be brief."

The evaluation data used in §4 and Appendix J consists of a separate set of 200 base queries. Although IFEval contains some length-related instructions, it offers limited examples with significant variation in length constraints and expressed in different ways (e.g., "I don't want anything longer

---

[6]https://openai.com/index/hello-gpt-4o/

Table 7: **Examples of Synthetically-generated Data.**

| Condition | Example |
|---|---|
| Original IFEval prompt (3 constraints) | Write a 300+ word summary of the wikipedia page https://en.wikipedia.org/wiki/Raymond_III,_Count_of_Tripoli. Do not use any commas and highlight at least 3 sections that have titles in markdown format, for example *highlighted section part 1*, *highlighted section part 2*, *highlighted section part 3*. |
| Synthetic single-constraint punctuation:no_comma | Write a summary of the wikipedia page https://en.wikipedia.org/wiki/Raymond_III,_Count_of_Tripoli. Do not use any commas. |
| Synthetic single-constraint detectable_format: number_highlighted_sections | Write a summary of the wikipedia page https://en.wikipedia.org/wiki/Raymond_III,_Count_of_Tripoli. Highlight at least 3 sections that have titles in markdown format, for example *highlighted section part 1*, *highlighted section part 2*, *highlighted section part 3*. |
| Synthetic single-constraint length_constraint:number_words | Write a 300+ word summary of the wikipedia page https://en.wikipedia.org/wiki/Raymond_III,_Count_of_Tripoli. |
| Synthetic no-constraint base query | Write a summary of the wikipedia page https://en.wikipedia.org/wiki/Raymond_III,_Count_of_Tripoli. |
| Synthetic augmented with detectable_format: number_bullet_lists | Write a 300+ word summary of the wikipedia page https://en.wikipedia.org/wiki/Raymond_III,_Count_of_Tripoli. Your answer should contain exactly 3 bullet points in markdown format. Use * to indicate bullets, like: * xyz * abc * opq |

than 30 words," or "Answer using five paragraphs"). We opt for synthetically generating data to enable consistent evaluation across a set of constraints varying over a narrower range (1 to 5 sentences).

**Word-specific Instructions.** For word-specific instructions, we compute vector representations for inclusion constraints, where the model is instructed to include a specific keyword in the output (e.g., "The output should contain the word $\{w\}$" or "Ensure that the word $\{w\}$ is included in the response"). For each keyword, we use a set of 20 base queries randomly sampled from those generated as described in Appendix C, ensuring none of the queries used for vector computation are reused in evaluation. The topics and content of these base queries are often semantically unrelated to the keyword being included. The same procedure is carried out for word-exclusion instructions in Appendix K.

For validation, we use GPT-4o to synthetically generate a set of questions similar to the base queries in IFEval. Additionally, the prompt requests the generation of a list of words likely to appear in the answer to each question. These question-word pairs (276 in total) are used for grid search validation of word inclusion and exclusion. For evaluation, we use IFEval examples containing keyword inclusion and exclusion instructions. Many examples contain multiple keyword constraints in a single query, so we separate these into individual prompts, each requesting the inclusion or exclusion of a single keyword. This process yields 86 evaluation prompts for keyword inclusion and 117 for keyword exclusion.

## E    STEERING LAYER AND WEIGHT SELECTION

When selecting the steering layer and weight, our goal is to identify a combination that improves instruction-following accuracy without compromising the fluency or comprehensiveness of the output. While instruction-following accuracy can be quantified through specific checks, evaluating fluency and comprehensiveness is more challenging, as thorough evaluations, such as our GPT-4o-based procedure, are computationally expensive and impractical for validation. To address this, we use perplexity computed by a smaller model as a proxy (GPT-2; Radford et al., 2019). The rationale is that very low perplexity values can indicate outputs where steering has significantly degraded quality. For each layer-weight combination, we compute the fraction of low-perplexity outputs (be-

Table 8: **Steering Layers for Format Instructions.** Layer indices used for steering across different models and evaluation settings (with and without explicit instructions) for each instruction in the format subset. A dash ("-") indicates that no steering is performed.

| Model Name | Phi-3 | | Gemma 2 2B IT | | Mistral 7B Instr. | | Gemma 2 9B IT | |
|---|---|---|---|---|---|---|---|---|
| Setting | w/o Instr. | w/ Instr. | w/o Instr. | w/ Instr. | w/o Instr. | w/ Instr. | w/o Instr. | w/ Instr. |
| Capital Word Freq. | 18 | 20 | 21 | 23 | 12 | 24 | 16 | - |
| Capitalize | 28 | 22 | 11 | 11 | 28 | 30 | 28 | 8 |
| Lowercase | 18 | 30 | 17 | 7 | 18 | 6 | 22 | - |
| Constrained Resp. | - | - | - | - | - | - | - | - |
| JSON Format | - | 6 | - | 13 | 16 | - | - | 10 |
| Multiple Sections | - | 8 | - | 9 | - | - | - | 28 |
| Number Bullet Lists | 20 | 16 | 5 | - | 24 | 12 | 16 | 12 |
| Highlighted Text | 26 | 18 | 5 | - | 24 | 26 | 12 | 10 |
| Title | - | - | - | - | - | - | 34 | - |
| No Comma | 10 | 12 | 11 | 5 | 14 | 26 | 22 | - |
| End Checker | - | 18 | - | 5 | - | 24 | - | - |
| Quotation | 26 | 24 | 11 | 23 | - | - | 34 | 12 |

Table 9: **Steering Layers for Language.** Layer indices used for steering across different models and evaluation settings (with and without explicit instructions) for each language-related instruction. A dash ("-") indicates that no steering is performed. In most cases with explicit text instructions, steering was unnecessary as the models typically followed the instruction and generated responses in the requested language.

| Model Name | Phi-3 | | Gemma 2 2B IT | | Mistral 7B Instr. | | Gemma 2 9B IT | |
|---|---|---|---|---|---|---|---|---|
| Setting | w/o Instr. | w/ Instr. | w/o Instr. | w/ Instr. | w/o Instr. | w/ Instr. | w/o Instr. | w/ Instr. |
| Language Ar | 18 | 16 | - | 15 | 22 | - | 20 | 20 |
| Language Bg | 22 | - | 19 | - | 18 | 6 | 22 | - |
| Language Bn | - | - | - | - | - | 18 | 20 | - |
| Language De | 16 | - | 15 | - | 16 | - | 20 | - |
| Language Fa | 20 | - | - | - | - | - | 22 | - |
| Language Fi | 20 | - | 15 | - | 20 | 12 | 22 | - |
| Language Gu | - | 14 | - | - | - | - | - | - |
| Language Hi | 22 | 14 | 15 | 7 | - | - | 22 | 12 |
| Language It | 16 | - | 15 | - | 18 | - | 20 | - |
| Language Mr | 30 | 14 | 17 | 15 | - | - | 24 | - |
| Language Pa | - | - | - | - | - | - | - | 8 |
| Language Pt | 16 | - | 15 | 7 | 16 | - | 20 | - |
| Language Ru | 16 | - | 15 | - | 18 | - | 22 | - |
| Language Sw | 20 | - | - | - | - | 24 | 22 | 20 |
| Language Ta | - | 6 | - | - | - | - | 22 | - |
| Language Te | - | - | - | 7 | - | - | 20 | - |
| Language Th | - | - | - | 5 | - | - | 20 | - |
| Language Ur | 28 | - | - | 15 | - | - | 28 | - |
| Language Vi | 20 | - | 15 | - | - | 12 | 20 | - |

low a threshold) and exclude configurations where this fraction is non-zero.[7] Among the remaining configurations, we select the one with the highest validation accuracy. Additionally, we compare the validation accuracy for each layer against a baseline without steering to ensure the intervention provides an improvement. If no layer improves upon the baseline, steering is not applied at test time. In cases where multiple layers achieve the same highest accuracy, we select the earliest layer. The perplexity threshold provides a mechanism for balancing instruction-following accuracy against the risk of quality degradation. A higher threshold results in a more conservative steering procedure but may exclude valid parameter combinations that could yield higher accuracy. In our experiments, we set the threshold to 2.5 for format instructions and 4 for word-specific instructions. This systematic approach ensures effective steering while maintaining output quality.

**Layer Selection Results.** In Table 8, we provide the layer indices used in our steering procedure for each instruction in the format subset (excluding language) across all models and evaluation settings (with and without explicit text instructions). We observe that for "Constrained Response" and "Ti-

---

[7]In non-instruction-tuned models low perplexity may occur even without steering. In these cases we exclude layers if the post-steering fraction exceeds the baseline value observed before steering.

Table 10: **Prompt for Quality Evaluation Question Generation.**

```
The following is a prompt that is used to evaluate the generations from a large
language model.  We do not know how to evaluate the quality of model answers for this
prompt.  Can you come up with 5 of less questions that can break down the quality to
simpler evaluation tasks that we can then ask about the model answer?  Each question
should have a simple yes, no answer.
 Prompt:  {{ prompt without instruction }}
List all sub questions in the following format:
Output:
1:  Question:  <question>
2:  Question:  <question>
...
N: Question:  <question>
```

Table 11: **Prompt for Quality Score Evaluation.**

```
We need to evaluate the quality of generations from a large language model.  You will
be given an input prompt, the response from a language model and a set of questions
assessing the quality of the response.  You need to review the response against the
input prompt and provide an answer to each question as either 'Yes', 'No' or 'Not
Applicable' if the question does not apply to the case along with a reason for your
answer.
 Prompt:  {{ prompt without instruction }}
 Response:  {{response}}
Questions:  {{evaluation questions}}
List your answers in the following format:
Output:
1.  Question:  <question>.  Reason:  <reason>:  Answer:  <answer>
2.  Question:  <question>.  Reason:  <reason>:  Answer:  <answer>
…
N. Question:  <question>.  Reason:  <reason>:  Answer:  <answer>
```

tle," virtually no layer improves performance with steering. The "Constrained Response" instruction asks the model to respond using only "yes," "no," or "maybe," while the "Title" instruction requires the answer to include a title wrapped in angular brackets (e.g., "$<>$"). In these cases, the steering vectors sometimes push the model too aggressively toward outputting instruction-related tokens, resulting in poor outputs (e.g., repeating characters like "$<<<<$"). For language instructions, the selected layers are listed in Table 9. For length instructions, we intervene at layer 12 in Phi-3 and layer 16 in Gemma 2 9B. For word-specific instructions, we use Phi-3's layer 24 and 28, and Gemma 2 2B IT's layers 24 and 22 for for inclusion and exclusion, respectively. The validation results used for layer and weight selection selection are reported in Appendix K).

**Candidate Weights.** We base the selection of the steering weight on the computation in Eq. (2). For format instructions, the weight from Eq. (2) is used directly (i.e., for each layer, there procedure above include a single layer-weight combination). For length and format instructions, the value computed in Eq. (2) is averaged across different inputs and is used as a reference to determine a range of suitable steering weights. For length instructions, we experiment with multiple weight values, highlighting the continuous nature of the constraint. For word inclusion instructions, we notice that in some cases (particularly with Gemma 2B), the weight is too low to impact the model's output. This could be because the steering vector is averaged across inputs regardless of whether the model satisfies the constraint. This issue tends to be more pronounced for word-specific constraints (than, e.g., for format instructions) since the strength of the instruction signal can vary greatly depending on whether the word is related to the base query (e.g., including a word related to the query is easier than including an unrelated one). To address this, in the selection procedure described above, we consider a small set of weights based on the value $\bar{c}$ (i.e., the value $c$ computed as in Eq. (2) averaged across different inputs). For Phi-3 $\bar{c} \approx 52$ at layer 28 and $\bar{c} \approx 42$ at layer 26, and we use the set $\{40, 60, 80, 100\}$ as grid-search value across the layers $\{24, 26, 28\}$. For Gemma 2 2B IT, $\bar{c} \approx 55$ at layer 22 and $\bar{c} \approx 74$ at layer 24, and we use the set $\{60, 80, 100, 120\}$ as grid-search value across the layers $\{22, 24\}$.

## F    RESPONSE QUALITY EVALUATION

### F.1    EVALUATION PROCEDURE

To evaluate the quality of the model generated responses, we first use the base queries outlined in Appendix C and prompt GPT-4o to produce 5 questions about the query to evaluate any model generated answer to the query (prompt reported in Table 10). For each model generated response in our evaluation, we then prompt GPT-4o to provide a binary "yes/no" answer to each question and "Not Applicable" if the question is not applicable to the query, along with a reason (prompt in Table 11). An aggregate quality score is computed as the average % of questions that is scored as "yes". We run the experiment three times for each model and report the quality score and standard error to account for any non-determinism from the GPT-4o based evaluator.

### F.2    EXAMPLES OF QUALITY SCORE CHANGES

While our evaluation procedure assesses how well the model's answers address the base queries, the method used to select steering locations and weights is based on IFeval's instruction-following accuracy metric. This metric consists of checks that verify specific aspects of the model's output related to instructions but sometimes these checks may be incomplete. For example, the default metric in IFEval for section-related formatting checks that the model includes sections in the generation but it does not check whether there is content in the section or the quality of the content if present. Therefore, in such cases this sometimes results in the selection of layer indices and steering weights that over-steer the model, leading to outputs that satisfy the instruction but have poor quality (e.g., repetitive token sequences). An example of this can be seen in the first row of Table 12. These issues are more prevalent in the Phi and Mistral models, compared to the Gemma models (noticeable from the smaller deltas for these models in Figure 4). This discrepancy may be due to Gemma's longer training and a better size/performance tradeoff, potentially making them more robust to activation steering.

Another source of quality drops can be minor factual inconsistencies in the model's responses. Steering may sometimes cause slight deviations from factual accuracy, which our quality score metric captures by incorporating GPT-4-generated questions about correctness. For example, in the second row of Table 12, the model inaccurately states that Kentucky, Missouri, and Mississippi were part of the original 13 colonies. While these inconsistencies can affect quality scores, they are not widespread and represent a general challenge for activation steering. Addressing factual consistency is difficult when selecting the steering location and weight, but it remains an important aspect to consider in future work. Finally, quality score drops can also result from small and arbitrary differences in the output, as shown in the third row of Table 12, in which the output looks reasonably good, but the model fails to follow the correct rhyme scheme for a limerick (AABBA), and the quality score drops by 0.6.

In conclusion, it is well-known that direct intervention on a model's activations can sometimes compromise the quality of the output. In our case, we observe relatively few instances where this happens, but improvements could be made. Future work could mitigate these issues by conducting a more exhaustive grid search over layers and steering weights, or by better modulating the steering intensity (Scalena et al., 2024; Stickland et al., 2024).

### F.3    ADDITIONAL RESULTS

We apply the response quality evaluation procedure to outputs generated by steering the model for length and word-specific instructions, following the same approach used for format constraints in §3. As in the previous section, we analyze the outputs produced by Phi-3, reporting the score changes due to steering both with and without explicit text instructions, as well as the quality score differences when text instructions are added without steering.

**Length Instructions.** Figure 9a presents the results for length constraints. We analyze a subset of 100 outputs from the generation process used for Figure 5b in §4 (which evaluated 200 examples). The model is steered with a weight of $c = 20$ on a set of base prompts, applying different length constraints requesting the model to answer using at most $n$ sentences ($n \in 1, \ldots, 5$). The two leftmost columns in Figure 9a show the changes in quality score when steering the model on inputs

Table 12: **Examples of Changes in Quality Score.** In the first case, the model is steered for the "Uppercase" instruction, which causes the output to become repetitive, resulting in a significant quality score drop from 1 to 0.4. In the second case, the model inaccurately states that Kentucky, Missouri, and Mississippi were part of the 13 colonies, leading to a drop in quality score from 0.8 to 0. In the third case, while the output looks fairly good, the model fails to follow the correct rhyme scheme for a limerick (AABBA), and the quality score drops from 0.8 to 0.2. These examples are taken from the top 10 outputs generated by Phi-3 with the largest drop in quality score.

| Query | Output w/o Steering | Output w/ Steering |
|---|---|---|
| The Legend of the Sword and the Fairy is a movie in which Wan Wan is a villain. Write a story about Wan Wan's character in list format. | 1. Introduction to Wan Wan:\n - Wan Wan is a cunning and ambitious antagonist in the movie "The Legend of the Sword and the Fairy."\n - She is a skilled sorceress with a deep-seated desire for power and control.\n 2. Early Life:\n - Wan Wan was born into a noble family, but her parents were killed in a tragic accident when she was young.\n - She was raised by her grandmother, who taught her the [...] | [...] 14. **WAN WAN'S REDEMPTION IS A MIRACLE**, AS HE AND THE SWORD AND THE FAIRY'S GIRL FIND HARMONY AND LOVE.\n 15. **WAN WAN'S REDEMPTION IS A TEACHING**, AS HE LEARNS THE VALUE OF LOVE, HONOR, AND SELFLESSNESS.\n 16. **WAN WAN'S REDEMPTION IS A MIRACLE**, AS HE AND THE SWORD AND THE FAIRY'S GIRL FIND HARMONY AND LOVE.\n 17. **WAN WAN'S REDEMPTION IS A TEACHING**, AS [...] |
| What is the difference between the 13 colonies and the other British colonies in North America? | The 13 colonies, also known as the Thirteen Colonies, were a group of British colonies on the Atlantic coast of North America founded in the 17th and 18th centuries that declared independence in 1776 and formed the United States. The other British colonies in North America, often referred to as the Middle and Southern colonies, were established later and had different characteristics. \n 1. Location: The 13 colonies were located along the Atlantic [...] \n 2. Economy: The 13 colonies [...] \n 3. Settlement Patterns: [...] | The 13 colonies, also known as the Thirteen Colonies, were a group [...] These colonies were: \1. Virginia \2. Massachusetts \3. New Hampshire \n 4. Rhode Island \5. Connecticut\n 6. New York\n 7. New Jersey\n 8. Pennsylvania\n 9. Maryland\n 10. Delaware\n 11. \n *Kentucky* (not originally a British colony, but later admitted as a state)\n 12. *Missouri* (not originally a British colony, but later admitted as a state)\n 13. *Mississippi* (not originally a British colony, but later admitted as a state) [...] |
| Write a limerick about writing a limerick. | Whose verses were both sharp and clean.\n He penned a limerick,\n With wit so quick,\n And the rhythm was quite the machine. | There once was a poet named Kent,\n Whose limericks were quite the event.\n He wrote one with glee,\n A limerick, you see,\n That was about writing a limerick, **a test.** |

with and without length instructions. These values are averaged over the 5 constraints, as the steering procedure applied is the same way for all 5 constraints (adding the *conciseness* vector with weight $c = 20$). A negative delta is expected, as steering the model to produce shorter responses generally reduces the comprehensiveness of the answer. This pattern is confirmed by the rightmost columns in Figure 9a, which represent the difference in quality score when length instructions are explicitly provided in the input text. The different colors correspond to the number of sentences ($n$), and as expected, shorter length constraints lead to larger drops in quality score. Although the effect of steering is not directly comparable to any specific length constraint (since steering applies an additional reduction in length without a one-to-one mapping between steering weight and sentence count), we observe that the changes in quality scores due to steering remain within a reasonable range when compared to those induced by text instructions.

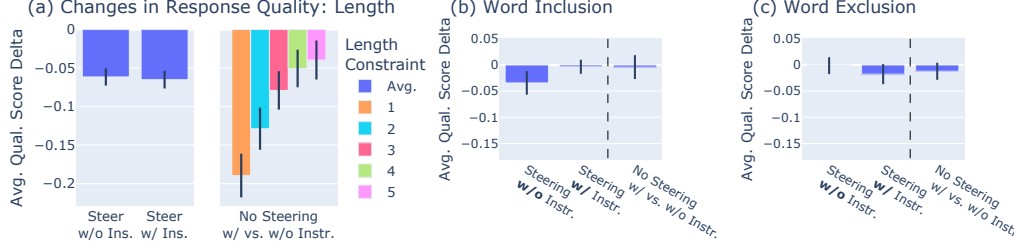

Figure 9: **Changes in the Response Quality Scores.** Quality score deltas for (a) length instructions, (b) word inclusion, and (c) word exclusion, across three conditions: steering without instructions, steering with instructions, and no steering with vs. without instructions.

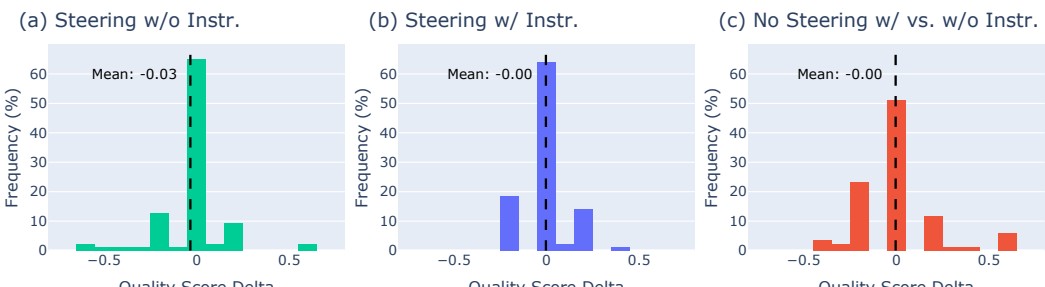

Figure 10: **Distribution of Quality Score Changes for Word Inclusion.** Histograms of quality score deltas under three conditions: (a) steering without explicit instructions, (b) steering with explicit instructions, and (c) no steering, comparing outputs with and without instructions. The distributions show minimal impact on quality scores across all settings, with mean values close to zero, with most shifts falling within the [-0.25, 0.25] range.

**Word-specific Instructions.** In Figures 9b and 9c, we report the changes in quality scores for instructions that request the inclusion or exclusion of a specific word in the answer.

The decrease in quality is minimal across all scenarios, the largest drop observed when steering for word inclusion without explicit instructions (around -0.03). We further investigate this by examining the empirical distributions of the score deltas for word inclusion instructions (Figure 10). The distributions appear centered around 0, with a few outliers showing deviations larger than 0.4. We perform a paired two-sided t-test to assess whether the means of these three distributions are significantly different. The p-values for the three tests (one for each pair) are all greater than 0.05 (0.25, 0.98, and 0.26). While the lack of significance does not prove that the distributions are identical, we interpret this as evidence of the minimal impact our steering procedure has on the model's ability to address the base queries.

**Format: Breakdown by Instruction.** Figure 11 presents the quality score deltas for format instructions, broken down by individual instructions across all four models (Phi-3, Gemma 2 2B IT, Mistral 7B IT, and Gemma 2 9B IT). The three settings compared are steering without explicit instructions, steering with explicit instructions, and no steering but comparing outputs with and without explicit instructions.

### F.4 QUALITY SCORE/ACCURACY TRADE-OFF

We investigate how the steering weight $c$ mediates the trade-off between instruction-following accuracy and output quality degradation for word exclusion constraints. Figure 12 shows the quality score and accuracy obtained by steering Phi-3 with varying negative values of $c$ (as we subtract the word-inclusion vector, as explained in §5 and Appendix J). In both settings (with and without explicit text instructions), larger absolute steering weights lead to higher instruction-following accuracy. This improvement comes at the cost of a gradual decrease in quality score, indicating a trade-off between strict adherence to constraints and maintaining output comprehensiveness. The

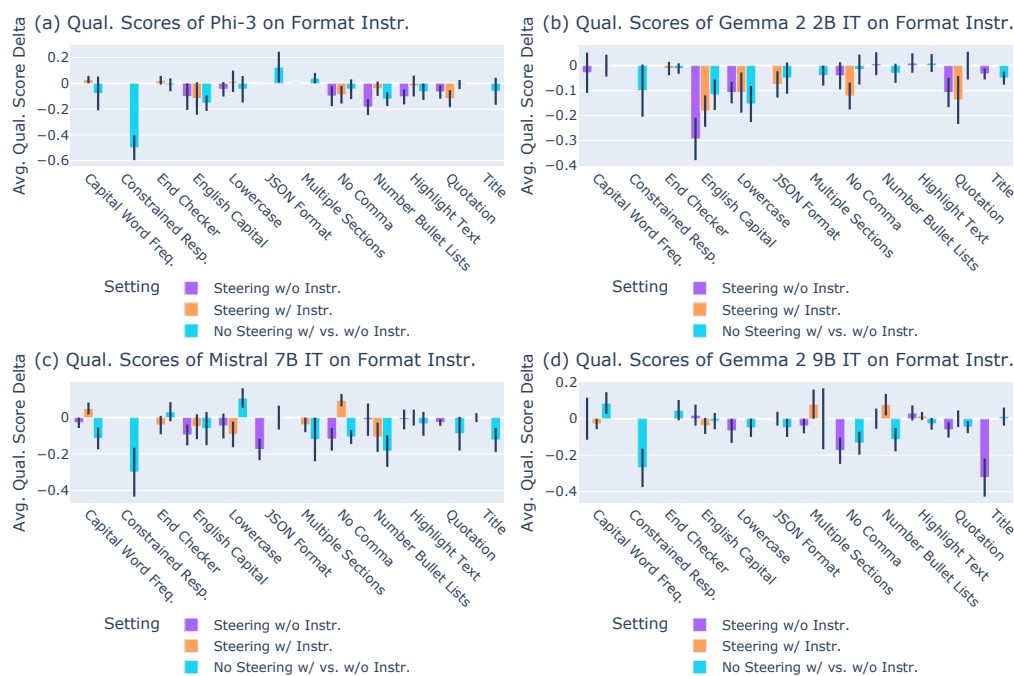

Figure 11: **Quality score deltas for format instructions across all models and settings.** (a) Phi-3, (b) Gemma 2 2B IT, (c) Mistral 7B IT, (d) Gemma 2 9B IT. Bars represent the average quality score delta under three conditions: steering without explicit instructions (purple), steering with explicit instructions (orange), and no steering but comparing outputs with and without explicit instructions (light blue).

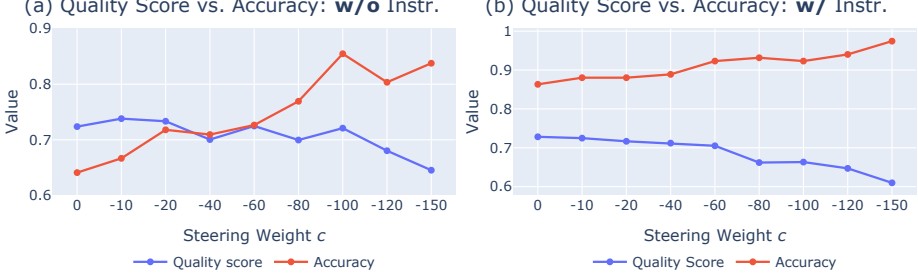

Figure 12: **Trade-off Between Quality Score and Accuracy on Word Exclusion.** (a) Results for steering without explicit text instructions, showing increasing accuracy at the cost of a gradual decrease in quality. (b) Results for steering with explicit instructions, demonstrating a similar trend with higher baseline accuracy.

smooth transitions in both metrics suggest that tuning $c$ offers fine-grained control over this balance, highlighting the flexibility of activation steering.

# G    IMPLEMENTATION AND EXPERIMENTAL DETAILS

**Tokenization and Decoding.** Steering vectors are computed at the last token of the input. For instruction-tuned models, this typically corresponds to the `<|assistant|>` token, which marks the transition from user input to the start of the model's generation.[8] For non-instruction-tuned models, we follow previous work (Kojima et al., 2022; Yang et al., 2024) and structure the prompt as "*Q:* {`problem`}\n*A:*" Model outputs are decoded greedily, with a maximum generation length

---

[8]https://huggingface.co/docs/transformers/main/en/chat_templating

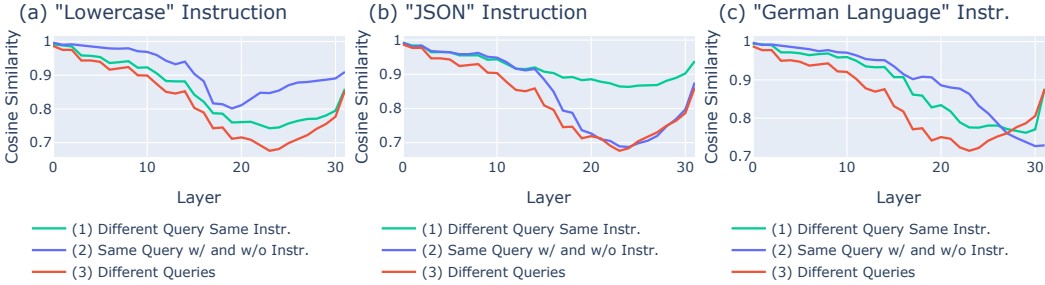

Figure 13: **Residual stream similarity across layers.** Cosine similarity of residual stream activations across layers for three instructions: (a) "Lowercase," (b) "JSON," and (c) "German Language." Comparisons are made between: (1) different queries with the same instruction (green), (2) same query with and without the instruction (blue), and (3) different queries without instruction (red). Results are shown for Phi-3 across all layers.

of 2048 tokens for format and length experiments, and 1024 tokens for keyword experiments. For efficiency, validation runs use a reduced maximum length of 384 tokens.

**Tools and Libraries.** For length-related experiments, word counts are measured as the number of space-separated sequences of characters and sentence counts are determined using NLTK (Bird & Loper, 2004). To assess whether the score improvement from steering is significantly different from standard inference, we carry out McNemar's test (McNemar, 1947). In particular, we use the exact version of the test, which uses the binomial distribution and is more conservative. The error bars reported in Figures 4 and 9 represent the standard error of the quality score computed over three different runs of GPT-4o. The smoothed empirical distributions shown in Figures Figure 5a, Figure 5c, Figure 8c, Figure 17a, and Figure 17c are obtained using kernel density estimation (Parzen, 1962). Our experiments were carried out using `PyTorch` (Paszke et al., 2019) and the `TransformersLens` library (Nanda & Bloom, 2022). We performed our data analysis using `NumPy` (Harris et al., 2020) and `Pandas` (Wes McKinney, 2010). Our figures were made using `Plotly` (Plotly Technologies Inc., 2015). The paper's bibliography was curated using `Ryanize-bib` (Zouhar, 2023).

## H ADDITIONAL RESULTS: REPRESENTATIONS

**Similarity of Activations.** Figure 13 shows the cosine similarity of residual stream activations in Phi-3 across different input sets. As described in §3, we compare the similarity of representations between: (1) pairs of inputs sharing the same base query, one with and one without the instruction; (2) inputs with different base queries but the same instruction; and (3) inputs with different base queries and no instruction. The similarity scores are reported across Phi-3's layers for three instructions: "lowercase" (no uppercase characters, Figure 13a), "JSON" (response formatted as JSON, Figure 13b), and "German Language" (Figure 13c).

**Geometry of Steering Vectors.** We perform a t-SNE dimensionality reduction on per-example instruction vectors, calculated as the difference in activations between inputs with and without explicit instructions, at layer 20 of Phi-3. The visualization for format instructions shows varying levels of cluster separation (Figure 14a). Specifically, instructions like "No Comma," "Lowercase," "JSON Format," "Title," and "Quotation" form distinct and well-separated clusters, indicating the model's ability to encode these constraints clearly. Instructions such as "Capitalize" and "Capital Word Frequency" cluster closely together, reflecting their similar semantic nature. In contrast, vectors for instructions like "End Checker" and "Constrained Response" are more dispersed, with no clear clustering, suggesting these instructions are less distinctly represented in the model's activation space.

Figure 14b presents a t-SNE dimensionality reduction of vectors computed for length instructions at layer 12 of Phi-3. Specifically, we include vectors derived from sentence-specific instructions (e.g., "answer using $n$ sentences," $n \in 1, \ldots, 5$) as well as more general "conciseness" and "verbosity" instructions, as described in §4. A clear linear trend emerges, with vectors for longer output in-

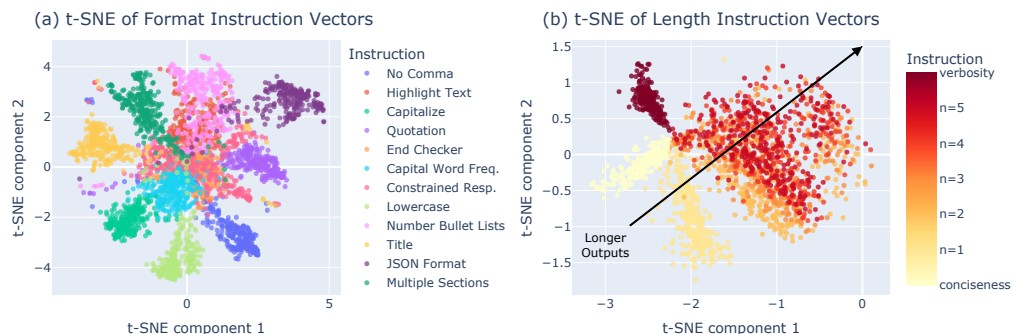

Figure 14: **t-SNE visualization of per-example instruction vectors.** (a) Format instruction vectors at layer 20 of Phi-3 for format instructions. Distinct clusters emerge for certain instructions (e.g., "No Comma," "Lowercase," "JSON Format"), while others (e.g., "End Checker," "Constrained Response") show less clear separation. (b) Vectors derived from length instructions at layer 12 of Phi-3. Vectors corresponding to concise outputs cluster on one end, while those for larger sentence counts ($n > 2$) cluster on the opposite end.

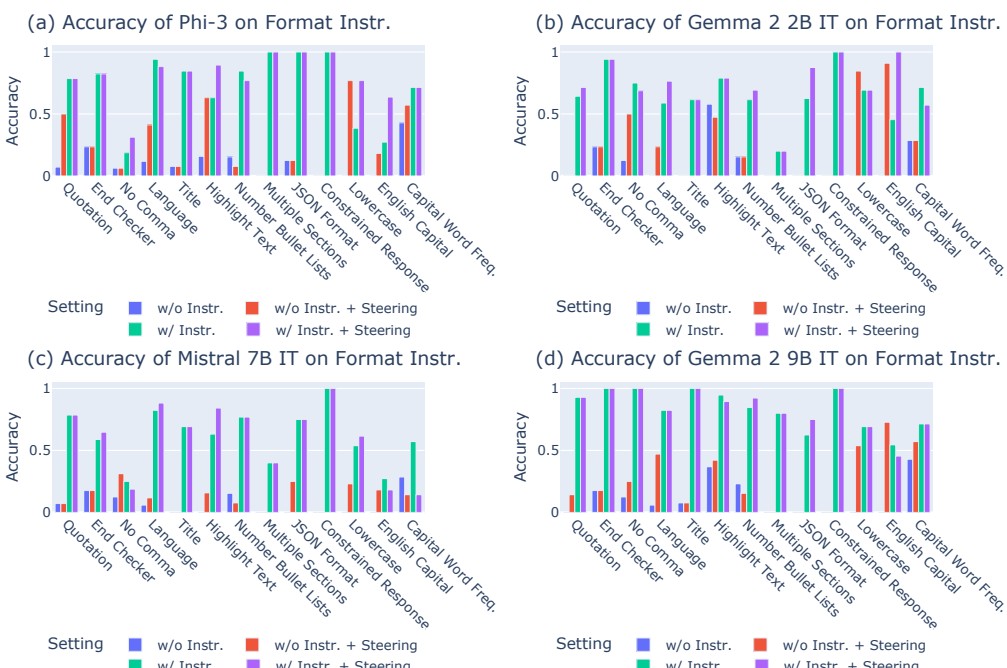

Figure 15: **Accuracy Breakdown by Format Instruction Across Models and Settings.** (a) Phi-3, (b) Gemma 2 2B IT, (c) Mistral 7B IT, and (d) Gemma 2 9B IT. Bars represent accuracy for format instructions under four settings: no explicit instruction (blue), steering without explicit instruction (red), explicit instruction without steering (green), and explicit instruction with steering (purple).

structions clustering at one end of the plot and those for concise outputs clustering at the opposite end.

# I ADDITIONAL RESULTS: FORMAT INSTRUCTIONS

**Validation Results.** Figure 16 provides examples of the results from the layer selection procedure. As detailed in Appendix E, we evaluate instruction-following accuracy and the fraction of low-perplexity outputs after steering at different layers. Panels (a) and (b) show examples obtained with Phi-3 in the no-instruction setting. In the first case ("Multiple Sections" instruction), the perplexity

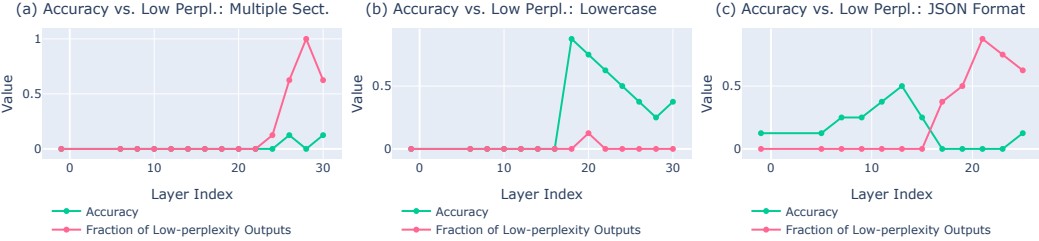

Figure 16: **Results from the layer selection procedure.** (a, b) Examples from Phi-3 in the no-instruction setting: (a) for "Multiple Sections," the perplexity check prevents selecting a layer with degraded output quality; (b) for "Lowercase," accuracy improves without quality loss. (c) For Gemma 2 2B IT with explicit "JSON Format" instructions, later layers lead to over-steering, with a degradation in both accuracy and quality.

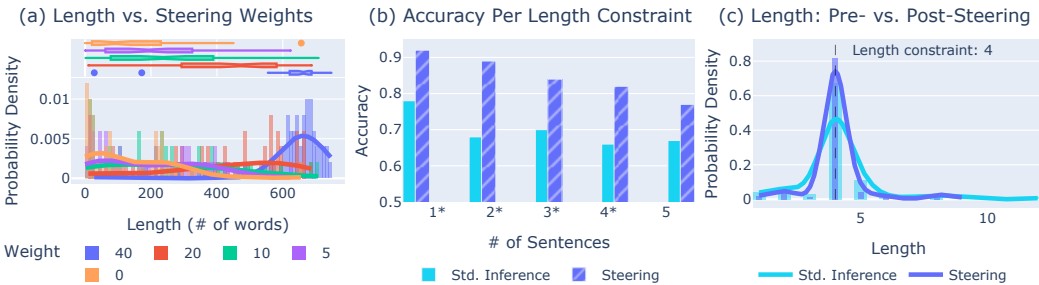

Figure 17: **Additional Results for Length Instructions.** (a) The distribution of output lengths when steering Phi-3 using a vector for verbosity, with varying steering weights. (b) Steering significantly improves accuracy for sentence-specific length constraints ($n \in 1, \ldots, 5$) in 4 of 5 cases. (c) For $n = 4$, steering shifts the output length distribution closer to the target.

check prevents the selection of a layer that would otherwise lead to a decrease in output quality, as indicated by a sharp rise in the fraction of low-perplexity outputs. Panel (c), which corresponds to the "JSON Format" instruction applied to Gemma 2 2B IT with explicit input instructions, illustrates a scenario where accuracy and output quality are correlated: steering at later layers degrades both quality and accuracy.

**Accuracy Breakdown by Instruction.** Figure 15 provides a breakdown of instruction-following accuracy across specific format instructions for all models and evaluation settings: no explicit instruction, steering without explicit instruction, explicit instruction without steering, and explicit instruction with steering. Across all models, steering improves accuracy for many instructions in the absence of explicit text instructions (blue vs. red bars). Notably, casing-related instructions ("Lowercase" and "English Capital") benefit strongly from steering, with Phi-3 and Gemma 2 2B IT achieving higher accuracy with steering alone than with explicit instructions (red vs. green bars). Certain instructions like "Title" and "Constrained Response" show near-perfect adherence with explicit instructions alone, leaving little room for improvement through steering.

## J    ADDITIONAL RESULTS: LENGTH INSTRUCTIONS

**Steering for Longer Outputs.** In Figure 17a, we show the distribution of output lengths when the model is steered using a verbosity vector. This vector is computed from inputs containing instructions to generate verbose responses (e.g., "Provide a long answer"). Similar to our procedure for the conciseness vector in §4, we apply different steering weights to modulate verbosity. Using the same 50 base prompts as in Figure 5a, the results show that higher steering weights result in longer outputs, demonstrating that activation steering can effectively control output length by adjusting a single scalar value.

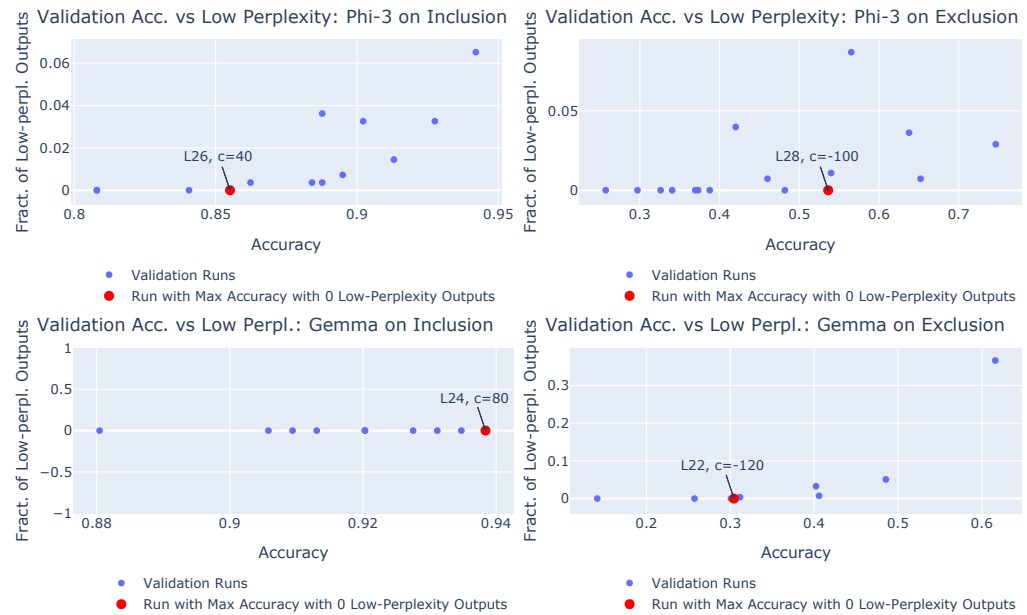

Figure 18: **Steering Layer and Weight Selection for Word-specific Instructions.** Each panel shows instruction-following accuracy (x-axis) vs. fraction of low-perplexity outputs (y-axis). Blue dots represent validation runs; red markers indicate the selected configuration with maximum accuracy and zero low-perplexity outputs. Results are shown for Phi-3 and Gemma 2 2B IT.

Table 13: **Examples of Tokens Promoted by Word-exclusion Vectors.** Projecting word-exclusion vectors onto the vocabulary results in cases where the excluded word is promoted. This suggests that word-exclusion vectors may not effectively steer the model toward avoiding these words. The vectors are computed at layer 30 of Phi-3.

| Word | Top Tokens |
|------|-----------|
| books | _books, _Books, books, _book, engl |
| congress | _Congress, _parlament, _Legisl |
| Europe | _European, _europé, Europe, _Europe |
| urbanization | _urban, urban, _Urban, _urb, _rural |
| chromosomes | chrom, _chrom, engl, _genom, _Zob |
| water vapor | _water, _Water, water, _agua |
| fish | _fish, _Fish, fish, _aqu, _marine |
| adult | _youth, _Youth, _Child, _Children |
| hypothesis | _instead, engl, _invece, _Instead |

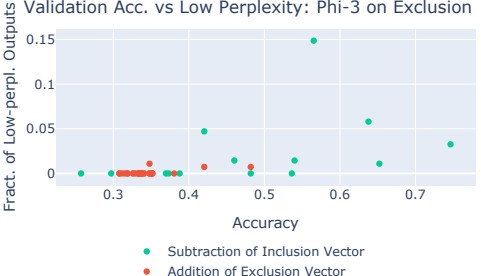

Figure 19: **Validation Accuracy vs. Perplexity Trade-off for Word Exclusion.** Subtracting word-inclusion vectors (green) outperforms adding word-exclusion vectors (red), showing the limitations of the latter.

**Steering for Exact Length.** We experiment with sentence-specific length instructions to evaluate the effectiveness of steering in improving adherence to explicit length constraints. Specifically, we compute steering vectors for instructions such as "Answer using $n$ sentences" ($n \in 1, \ldots, 5$) and apply them to steer the model toward satisfying the corresponding constraints. In these experiments, we use Phi-3 and the same combination of steering layer and weight as in §4 (layer 12, $c = 20$). The results in Figure 17b show that steering significantly improves the model's adherence to the specified lengths when explicit instructions are present in the input. Notably, steering produces statistically significant improvements in 4 out of the 5 cases. Figure 17c illustrates the effect of steering on the distribution of output lengths for a specific constraint ($n = 4$ sentences). Steering shifts the distribution closer to the target length, with a sharper peak at the desired value, while still maintaining some natural variability.

## K   ADDITIONAL RESULTS: WORD-SPECIFIC INSTRUCTIONS

**Validation Results.** As described in Appendix E, to determine the optimal steering layer and weight parameter for each model-setting combination, we validate our approach by jointly considering instruction-following accuracy and the fraction of low-perplexity outputs. The x-axis in each panel of Figure 18 represents the instruction-following accuracy, while the y-axis shows the fraction of low-perplexity outputs, which we aim to minimize. For each model and setting, we select the layer and weight combination that achieves the highest accuracy, provided the fraction of low-perplexity outputs is zero. Red markers indicate the chosen configurations.

**Steering using Word-exclusion Vectors.** Following our approach for word inclusion, we compute instruction vectors for word exclusion. However, projecting these exclusion vectors onto the model's vocabulary via the unembedding matrix often results in high logit values for tokens corresponding to the word that should be excluded. Examples of this phenomenon are shown in Table 13. While this effect does not occur in all cases (e.g., for words like "adult" and "hypothesis"), the presence of a positive signal for the excluded word in the logit space suggests that these vectors may counteract the intended steering effect. To confirm this, we compare the performance of steering using word-exclusion vectors versus subtracting word-inclusion vectors for the same words. As shown in Figure 19, subtracting word-inclusion vectors consistently yields higher accuracy and achieves a better accuracy-to-low-perplexity trade-off than adding word-exclusion vectors. These results indicate that subtracting inclusion vectors is a more effective and reliable method for steering the model to avoid specific words.

