# OpenReview forum: "Improving Instruction-Following in Language Models through Activation Steering"
_ICLR.cc/2025/Conference — ICLR 2025 Poster_

### Official Review · Reviewer_XmmQ · 2024-10-31

**Soundness:** 3
**Presentation:** 3
**Contribution:** 3
**Rating:** 8
**Confidence:** 4

**Summary:**

This paper develops a way to obtain steering vectors from  instruction prompts. Focusing on output modifiers such as “answer in uppercase,” they develop a way to obtain steering vectors by contrasting instructions with and without the modifier and averaging representation vectors from many paired samples, then they propose a steering approach that introduces scaling to the mean target magnitude.  They gather 12 format modifiers and 19 language modifiers and test their method on a data set of instructions on four language models ranging from about 3b to 9b parameters, sweeping over layers.  They test the ability to use the steering alone to add modifiers, and to use it to strengthen a modifier in a prompt. They also test vector negation by reversing word-inclusion modifiers to become word-exclusion, and they test composition of modifier vectors.  They also find that vectors from instruction-tuned models work very well on original models in some cases.

**Strengths:**

The derivation of steering vectors from instructions is an interesting research target and a good extension beyond similar work deriving such vectors from binary states or ICL prompts. The paper’s choice of deriving steering vectors from instruction modifiers is clever and novel, which allows the authors to create diverse training sets and also easily quantify the accuracy of results. The paper investigates nearly all the natural applications with useful measurements, including measurements of quality degradation in the presence of steering.  Positive results on negation and composition are interesting to see, and it is particularly interesting to see that vectors derived from instruction-tuned models work better than vectors from pretrained models when applied to pretrained models.

**Weaknesses:**

The presentation was sometimes uneven, giving the impression that the results might be curated to avoid showing results that did not work very well. Such cherry-picking should be resisted. For example: Figure 3 slices efficacy data by model and by instruction in different ways but does not provide breakouts by task for all four cases (bare prompt, prompt+modifier, prompt+steer, prompt+modifier+steer), which would help the reader build intuition about the failure cases.

Similarly: quality degradation was measured differently for different tasks, making it hard to compare. The paper would be improved if the appendix plotted or had a uniform table of quality degradation, computed the same way for every comparable task.

Other failure cases are mentioned but not measured: word exclusion is described as unpromising due to the presence of an embedding signal, but measurements of its failure are not shown. Full results should be shown.

It is stated that “it seems impractical to compute a separate steering vector for each possible length,” however that doesn’t seem impractical at all. Such specific-length steering vectors should be computed and compared to one another, and also to the conciseness concept described in the paper. If they do not work, that should be quantified and shown.

In appendix tables 8 and 9, many configurations are omitted with the explanation that "steering was unnecessary as the models typically followed the instruction."  Again, negative or "unnecessary" results should not be omitted. A key goal should be to explore and explain the limits of the observed effects. Failure or unhelpful cases are an important subject of experiments.

Some natural questions are unanswered, for example, whether the clusters of steering vectors over which means are taken are cleanly separated from each other (i.e., before taking means) or not.  It would be informative to plot a projection of the raw steering vectors for several of the tasks, for example, in a scatterplot as done in Hendel 2023.  In particular it would be interesting to see how closely-related vectors such as “answer of length n” for various n are arranged with respect to each other in representation space.

The choice of “c” is not fully justified, and it seems that the scaling factor “c” might be arbitrary. For example, c might mediate a tradeoff between efficacy and quality degradation.  It would be informative to plot tradeoffs over a sweep of c, if that is the case.

**Questions:**

* Can the the full grid of performance measurements of every modifier task (including both format and language) in all four cases (bare prompt, prompt+modifier, prompt+steer, prompt+modifier+steer) across models be included, perhaps in appendix?
* Similarly, quality degradation is broken out for some settings but not others. Can quality degradation measurements be shared in the same way over all modifiers?
* What are the measured results of word exclusion prompt steering vectors and specific numeric length modifiers? If they are failure cases, it will be helpful to see the extent to which they fail, and to share some of the typical behavior.
* Do the populations of sampled vector differences separate cleanly for different tasks, or is there some overlap?  It will be informative to plot these.
* Does the scaling factor induce a smooth tradeoff between output quality and accuracy in following the instruction?  It would be helpful o measure this.

---

> ### Author Response · Authors · 2024-11-21
>
> Thank you for recognizing our work’s “interesting research target” and  “clever and novel” contributions. We greatly appreciate your thoughtful feedback and suggestions, which have prompted additional insights and improvements to our paper.
>
> > **Weakness 1**: Figure 3 [...] does not provide breakouts by task for all four cases. & **Q1**: Can the the full grid of performance measurements of every modifier task [...] across models be included, perhaps in appendix?
>
> Yes, we have added a bar plot (Figure 15 in Appendix I) that provides a full breakdown of instruction-following accuracy for format instructions across all four cases (bare prompt, prompt + modifier, prompt + steering, prompt + modifier + steering) and all models.
>
> > **Weakness 2**: quality degradation was measured differently for different tasks [...] The paper would be improved if the appendix plotted or had a uniform table of quality degradation, computed the same way for every comparable task. & **Q2**: Can quality degradation measurements be shared in the same way over all modifiers?”
>
> We would like to point out that quality degradation was actually measured in the same way across tasks using the same GPT-4o-based procedure described in Section 2.3. The metric quantifies quality as the difference in response quality scores between two conditions. In particular, in Figures 4, 9a, 9b, and 9c, the y-axis represents this quality score delta. However, to further address your point, we have added a detailed breakdown of quality score deltas across all format instructions, tasks, and models in Figure 11 (Appendix F.3).
>
> > **Weakness 3**: word exclusion is described as unpromising due to the presence of an embedding signal, but measurements of its failure are not shown. & **Q3.1**: What are the measured results of word exclusion prompt steering vectors [...]?
>
> We included in the paper additional results to better illustrate the limitations of word-exclusion steering (Appendix K). In particular:
> - Results showing that word-exclusion vectors often promote the excluded word or its sub-tokens in the vocabulary space, confirming the presence of an embedding signal that counteracts the desired steering behavior (Table 13).
> - Validation experiments comparing the addition of word-exclusion vectors to the subtraction of word-inclusion vectors demonstrating that the latter is more effective (Figure 19). These results highlight the limitations of direct, out-of-the-box word-exclusion steering.
>
> > **Weakness 4**: specific-length steering vectors should be computed and compared to one another, and also to the conciseness concept described in the paper.  & **Q3.2**: What are the measured results of [...] specific numeric length modifiers?
>
> The reason why we deemed it impractical to compute steering vectors that capture exact length constraints is the variability in how such constraints can be expressed (e.g., as number of characters, words, sentences, lines, or paragraphs), which lacks a clear correspondence. For this reason, we opted for capturing more generally applicable steering concepts such as conciseness and verbosity, which, regardless of the input or the explicit text instruction, could adjust the length of the generation by making it longer or shorter.
>
> However, based on your feedback, we extended our experiments to include specific-length instructions. In particular, we computed steering vectors for sentence-specific length constraints (e.g., “Answer using $n$ sentences” for $n \in {1, …, 5}$). The results, presented in Figures 17b and 17c, demonstrate that steering with these vectors significantly improves the model’s adherence to the specified length constraints. For Phi-3, steering yields statistically significant improvements in 4 out of 5 cases.
>
> Thank you for the suggestion, which prompted additional experiments to further validate the effectiveness of our steering approach.
>
>
> [continues in the next comment]

---

> ### Author Response · Authors · 2024-11-21
>
> > **Weakness 5**: In appendix tables 8 and 9, many configurations are omitted with the explanation that "steering was unnecessary as the models typically followed the instruction." Again, negative or "unnecessary" results should not be omitted.
>
> We believe this may be a misunderstanding. In Tables 8 and 9 there are no configurations that are omitted: the tables explicitly report layer indices for steering or indicate with a dash (“-”) that no steering was performed. The dash indicates cases where the steering layer selection did not register any accuracy improvements, making steering unnecessary. For example, some language instructions are followed accurately without steering when explicit instructions are present, and no layer-weight combination improves performance.
>
> > **Weakness 6**: It would be informative to plot a projection of the raw steering vectors for several of the tasks, for example, in a scatterplot as done in Hendel 2023. In particular it would be interesting to see how closely-related vectors such as “answer of length n” for various n are arranged with respect to each other in representation space.
>
> Thank you for this insightful suggestion. Following Hendel et al., 2023 [1], we performed t-SNE dimensionality reduction on per-example steering vectors (computed as the activation difference between inputs with and without instructions):
> - For format instructions (Figure 14a), we observe well-separated clusters for instructions such as “No Comma,” “Lowercase,” and “JSON Format,” indicating distinct representations. Related instructions like “Capitalize” and “Capital Word Frequency” are closer in representation space, reflecting their semantic similarity. Scattered clusters, such as “End Checker,” suggest less consistent representations.
> - For length instructions (Figure 14b), we observe a clear linear trend: shorter-output instructions cluster at one end, while verbosity and longer-output instructions lie towards the opposite end. This interesting observation reflects the continuous nature of length-related constraints.
>
> > **Weakness 7**: c might mediate a tradeoff between efficacy and quality degradation. It would be informative to plot tradeoffs over a sweep of c, if that is the case. & **Q6**: Does the scaling factor induce a smooth tradeoff between output quality and accuracy in following the instruction?
>
> To address this, we conducted a sweep over the steering weight $c$ and analyzed the trade-off between instruction-following accuracy and quality degradation. We added the results to the paper in Figure 12, which shows that larger absolute values of $c$ lead to higher accuracy, but this comes at the cost of a gradual decline in quality score. This smooth trade-off is consistent across both settings (with and without explicit instructions).
> When computing validation scores during layer and weight selection, we approximate this trade-off using the fraction of outputs with extremely low perplexity as a proxy for quality degradation. This approach allows us to systematically select $c$ values that balance accuracy and quality.
>
> ---
>
> In conclusion, we would like to thank the reviewer for the many valuable suggestions, which have led to the inclusion of several additional results and insightful analyses in our paper. We hope that these enhancements, along with the improved accuracy/quality trade-off demonstrated in our updated findings, will be reflected in the reviewer’s final evaluation of our work.
>
> ---
>
> [1] Hendel, R., Geva, M. and Globerson, A., 2023. In-context learning creates task vectors. arXiv preprint arXiv:2310.15916.

---

> > ### Comment · Reviewer_XmmQ · 2024-11-25
> > **More complete analysis makes the paper more solid**
> >
> > Thanks for adding the various extra detailed empirical results in the appendix - these will be helpful for researchers who are building upon your work. They help give intuition for where the approach works best (and where not).
> >
> > I have raised my review score to reflect the improvements.

---

> > > ### Author Response · Authors · 2024-11-25
> > >
> > > Your positive feedback is greatly appreciated. Thank you!

---

### Official Review · Reviewer_B97K · 2024-11-01

**Soundness:** 3
**Presentation:** 4
**Contribution:** 2
**Rating:** 8
**Confidence:** 4

**Summary:**

This paper applies an activation steering method to improve the constraint following of LLMs such as response length and format. It also shows that multiple steering can be applied simultaneously and some steering generalizes from the instruct model to its base version.

**Strengths:**

- The steerability of LLMs, which this is tackling, is an important research area that has real-world implications.
- There are interesting findings in the paper, such as combining steering vectors and generalization to the base model.
- The paper is well written and easy to understand. The experimental setups look solid and diverse.

**Weaknesses:**

- Novelty: the method of steering by activation itself is not novel and has been used in other papers. The authors claim the application is different, but previous works on writing style and topic changing are related to response format and mentioning a word. Some aspects of the method (combining steering vectors) seems novel, but maybe the author can clarify exactly which parts are novel.
- Motivation: From reading the paper, the motivation was not that very clear. Why is this method necessary? Explicit instruction in words seems to work better and simpler, while the proposed method can lead to nonsensical responses. What is the main motivation of using this method instead?
- The proposed method doesn’t seem very general and may require adjusting for each instruction. For example, the method led to an opposite effect in the word exclusion task. In the length task, it is not clear how the exact length is translated into the vector scaling parameter.
- The dev set is using the same base queries as the evaluation? This makes it hard to judge how the method will generalize to unseen queries.
- It seems some task have extremely low number of samples. For example, there are only 12 sample for the format task, is that correct?
- There are some typos around L175 where commas are placed incorrectly.

**Questions:**

- Maintaining quality is important for steering methods, but only one of the experiments shows the quality evaluations. I saw some results in the appendix, but I think it needs to be discussed in the main text.
- In the length experiment, c=20 steering seems to help with different length constraints. I want to know if this steering makes the response generally shorter, which then happens to increase accuracy, or it makes it adhere to the specific length instruction more strongly? Because of the fixed c=20 value, I feel like it is likely the former.
- The plots are too small and sometimes use similar colors (dark blue vs light blue), which makes them hard to see. I think the writing can be made more concise to make more room for the plots.
- How are the base models following instruction? Is few-shot prompting used?
- Seems to be missing this related work that also does “capitilzation” https://arxiv.org/pdf/2311.06668

---

> ### Author Response · Authors · 2024-11-21
>
> Thank you for recognizing that our work tackles “an important research area that has real-world implications” with a “solid and diverse” experimental setup. We appreciate your thoughtful comments and constructive suggestions.
>
> > **Weakness 1**: Some aspects of the method (combining steering vectors) seems novel, but maybe the author can clarify exactly which parts are novel.
>
> We identify the primary novel contributions of our work as follows:
> 1. While activation steering has been used for controlling high-level concepts such as sentiment, style, and safety in prior work, our contribution lies in adapting this method to lower-level, fine-grained constraints defined through natural language instructions. For example, modifying writing style (e.g., formal to informal writing in Liu et al. [1], or negative to positive sentiment in Tigges et al. [2]) is different than controlling for specific fine-grained constraints on the format of the output such as “the response should be valid JSON” or excluding a specific word from the output. These tasks demand more precise control of the model’s output, which is fundamentally different from modifying broad stylistic features such as tone or sentiment.
> 2. As the reviewer noted, another novel aspect of our work is the composition of steering vectors, which enables simultaneous adherence to multiple constraints (e.g., combining instructions for format and length). This capability adds versatility to the method, addressing real-world use cases where multiple constraints often coexist.
> 3. We are also the first to demonstrate that cross-model steering—using vectors derived from instruction-tuned models—can outperform same-model steering in base models. This suggests a potential for leveraging specialized representations from fine-tuned models to steer base models effectively, enabling composable transfer learning in instruction-based tasks.
>
> > **Weakness 2**: What is the main motivation of using this method instead?
>
> Our method offers practical advantages while also contributing to a deeper understanding of how language models represent instructions. One significant benefit is the performance improvement observed even when explicit instructions are provided in the input. Our steering procedure enhances instruction adherence by correcting instances where models fail to follow instructions effectively. This highlights the method’s ability to reinforce instruction-following behavior in scenarios where models might otherwise exhibit inconsistent or incomplete adherence. Additionally, our work is driven by the motivation to understand how models internally represent instructions and whether these representations can be extracted and manipulated to influence behavior.
> Experiments in the “no instructions” condition provide a clean, controlled setup to demonstrate that the extracted representations successfully capture the instruction of interest.
>
> A second practical advantage of our method is the potential to reduce computational overhead.
> Results in the “no instructions” setting show that we can control the model’s output in specific ways without explicit instructions in the prompt. The computational overhead of the forward passes needed to process the instruction tokens can be significant, especially for large models and for long instructions.
> Since our steering procedure has a negligible impact on the computational cost (simple vector addition per forward pass), our method could be used to reduce the computation cost at inference time.
>
> > **Weakness 3**: The proposed method doesn’t seem very general and may require adjusting for each instruction.
>
> We acknowledge that real-world deployment of this method may require tailoring for specific constraints. However, part of our contribution is precisely to explore how activation steering can be adapted to diverse types of instructions. The set of instructions we study, which includes format, length, and keyword constraints, represents a range of practically relevant use cases. Additionally, many real-world scenarios involve combinations of instructions (e.g., “write the response in JSON and include no more than 5 sentences”). Our experiments in Section 6 demonstrate that steering can effectively handle such multi-instruction settings, suggesting the method’s utility in addressing complex user requirements. In practice, efforts for scaling up improvements in instruction-following may require learning a library of such representation that can be effectively reused for scenarios of interest.
> Regarding the question about exact-length constraints: We have added new results showing that steering increases adherence to exact-length instructions (e.g., “answer using exactly 4 sentences”), even when these constraints are explicitly stated in the input (Figure 16 in Appendix J).
>
> [continues in the next comment]

---

> ### Author Response · Authors · 2024-11-21
>
> > **Weakness 4**: The dev set is using the same base queries as the evaluation?
>
> No, the dev set consists of base query-instruction combinations that are entirely disjoint from those in the test set. Specifically, for each instruction, the base queries used to compute and validate steering vectors do not overlap with those used during evaluation.
>
> > **Weakness 5**: There are only 12 sample for the format task, is that correct?
>
> No, our experiments on format instructions use 163 examples for evaluation. Additionally, we conduct statistical significance tests that account for the size of the evaluation sets.
>
> > **Weakness 6**: There are some typos around L175 where commas are placed incorrectly.
>
> We believe this comment refers to the placement of commas relative to quotation marks. Following standard American English notation, commas and periods are placed inside quotation marks.
>
> > **Q1**: Maintaining quality is important for steering methods, but only one of the experiments shows the quality evaluations. I saw some results in the appendix, but I think it needs to be discussed in the main text.
>
> We would like to clarify that we evaluate output quality across all three instruction categories (format, length, and keyword instructions), as shown in Figures 4, 9a, and 9c, respectively.
> Nonetheless, we do agree on the importance of quality evaluation and, as mentioned in the general response, have improved our steering procedure by incorporating a perplexity-based quality check during layer and weight selection. This additional step ensures that steering parameters are chosen to maximize instruction-following accuracy while minimizing quality degradation. This updated approach demonstrates a reduction in quality score drops while retaining most of the gains in instruction-following accuracy, resulting in a more effective trade-off.
>
>
> > **Q2**: I want to know if this steering makes the response generally shorter, which then happens to increase accuracy, or it makes it adhere to the specific length instruction more strongly? Because of the fixed c=20 value, I feel like it is likely the former.
>
> Correct, steering for conciseness with steering weight c=20 leads to generally shorter answers. This effect is noticeable in length distributions shown in Figure 3c, where the post-steering length distribution where the post-steering distribution shifts left.
> However, as mentioned in our response to Weakness 3, we have added experiments that show steering vectors improve adherence to specific length instructions (e.g., “answer using exactly 4 sentences”).
>
> > **Q3**: The plots are too small and sometimes use similar colors (dark blue vs light blue), which makes them hard to see. I think the writing can be made more concise to make more room for the plots.
>
> We appreciate the reviewer’s feedback. During this rebuttal, our focus has been on addressing the reviewers’ questions and performing the requested analyses. We have not yet revisited the graphic presentation of the paper, but we will incorporate these suggestions in the next version, improving plot size, color distinctions, and adjusting the text to allow more room for visuals.
>
>
> [continues in the next comment]

---

> ### Author Response · Authors · 2024-11-21
>
> > **Q4**: How are the base models following instruction? Is few-shot prompting used?
>
> All experiments are conducted in a zero-shot setting. For base models, we follow prior work [3, 4] and structure prompts in the format: “Q: {problem}\nA:.”
>
> > **Q5**: Seems to be missing this related work that also does “capitilzation” https://arxiv.org/pdf/2311.06668
>
> Thank you for pointing this out. We already cite this work (Liu et al., 2024 [1]) in our paper. In their study, the authors compute vector representations for tasks described using a few in-context examples and apply these vectors to steer models toward goals such as dialogue safety and altering the style or tone of text. A key distinction from our work is that we do not rely on annotated few-shot examples to compute instruction representations. Instead, we derive these representations by contrasting activations from paired inputs with and without instructions, enabling a modular and scalable approach to instruction-following tasks.
>
> ---
>
> In conclusion, we would like to thank the reviewer for their detailed and constructive feedback. We hope that our responses have addressed their concerns and provided sufficient clarification.
>
> ---
>
>
> [1] Liu, S., Ye, H., Xing, L. and Zou, J.Y., In-context Vectors: Making In Context Learning More Effective and Controllable Through Latent Space Steering. In Forty-first International Conference on Machine Learning.
> [2] Tigges, C., Hollinsworth, O.J., Geiger, A. and Nanda, N., 2023. Linear representations of sentiment in large language models. arXiv preprint arXiv:2310.15154.
> [3] Kojima, T., Gu, S.S., Reid, M., Matsuo, Y. and Iwasawa, Y., 2022. Large language models are zero-shot reasoners. Advances in neural information processing systems, 35, pp.22199-22213.
> [4] Yang, C., Wang, X., Lu, Y., Liu, H., Le, Q.V., Zhou, D. and Chen, X., Large Language Models as Optimizers. In The Twelfth International Conference on Learning Representations.

---

> ### Author Response · Authors · 2024-12-01
>
> As the discussion period is approaching its conclusion, we would like to thank the reviewer once again for their feedback, which has significantly contributed to improving our paper. During the rebuttal, we incorporated several of the reviewers’ suggestions, adding new results and analyses that addressed their points. This led to stronger results and 6 additional pages of content. We kindly ask if the reviewer is satisfied with our responses and whether they would consider revisiting their scores to reflect these improvements.

---

> > ### Comment · Reviewer_B97K · 2024-12-03
> >
> > Thanks for addressing my questions. I think the results are solid and there may have applications, so I have raised my score.

---

> > > ### Author Response · Authors · 2024-12-03
> > >
> > > Thank you for taking the time to review our updates and for raising your score. We greatly appreciate your feedback and support!

---

### Official Review · Reviewer_R96q · 2024-11-02

**Soundness:** 3
**Presentation:** 3
**Contribution:** 3
**Rating:** 6
**Confidence:** 3

**Summary:**

The paper "Improving Instruction-Following in Language Models Through Activation Steering" proposes using activation steering to enhance the instruction-following abilities of language models. By deriving instruction-specific vectors based on activation differences between inputs with and without instructions, the method adjusts model behavior during inference to meet specific constraints such as output format, length, and keyword control.

**Strengths:**

The paper introduces a novel approach to activation steering through instruction-specific vector representations, allowing dynamic control over model behavior without retraining.

The paper covers multiple constraint types (format, length, word-specific) across several models, highlighting the robustness and adaptability of the approach. The cross-model transferability experiments are valuable, showing that steering vectors from instruction-tuned models improve base models.

The paper is well-structured and clearly explains each step.

Significance: activation steering offers a scalable solution for fine-grained control over LLM outputs. The demonstrated cross-model transferability suggests a cost-effective way to get instruction-following improvements.

In sum, this paper presents an original, well-executed, and practical contribution, for various real-world applications.

**Weaknesses:**

Cross-model transferability is a promising aspect of this work, yet the analysis here could benefit from more quantitative depth. Specifically, it would be useful to test how much performance degrades when steering vectors from instruction-tuned models are applied to base models of different architectures or parameter sizes.

The paper notes minor drops in response quality when adhering to certain constraints, particularly in the length and word-specific steering tasks. While the authors discuss this, it would be beneficial to implement strategies to mitigate these effects.

**Questions:**

The need for unique vectors per instruction could impact scalability, particularly for highly variable, user-specific instructions. Clarifying how the approach handles diverse variable instructions would add practical insights.

---

> ### Author Response · Authors · 2024-11-21
>
> Thank you for recognizing that our work represents an “original, well-executed, and practical contribution for various real-world applications.” We address your points below.
>
> > **Weakness 1**: Cross-model transferability is a promising aspect of this work [...] It would be useful to test how much performance degrades when steering vectors from instruction-tuned models are applied to base models of different architectures or parameter sizes.
>
> Thank you for pointing this out. We agree that cross-model transferability is a promising avenue and appreciate your suggestion for deeper quantitative analysis. Since two models that do not share the architecture and part of the pre-training will have completely different representation spaces, we expect the transfer of steering vectors across models to require a transformation to be effective. For example, a learned projection matrix could map residual stream representations from model A to the corresponding representations in model B, enabling steering vector transfer.
> While this exploration is beyond the current scope, it is an exciting direction for future work. Such an approach could allow for broader applicability for steering techniques and better understanding of cross-model representations that can lead to skill transfer, even across heterogeneous model families. We have added a discussion of this direction to the paper in Appendix A.
>
> > **Weakness 2**: The paper notes minor drops in response quality when adhering to certain constraints [...] While the authors discuss this, it would be beneficial to implement strategies to mitigate these effects.
>
> Minor drops in response quality are expected, particularly for constraints that inherently limit the comprehensiveness of the response (e.g., it is inevitable that a 2-sentence response might be less comprehensive than a 5-sentence response). However, we do agree that maintaining response quality is an important consideration in the design of the steering procedure. To address this, we have implemented a validation-time quality check that considers perplexity as a proxy for fluency and comprehensibility. This approach allows us to select steering parameters (layer and weight) that maximize instruction-following accuracy while minimizing quality degradation. As described in the general response, the updated results in the paper demonstrate that this approach effectively preserves most of the gains in instruction-following accuracy while significantly reducing the drop in quality score, achieving a better trade-off between accuracy and quality. Details of this procedure are provided in Appendix E, and we updated Figures 3, 4, 6, 7, and 8 in the main paper, as well as Figure 9 in the appendix to reflect these improvements.
>
> > **Q1**: Clarifying how the approach handles diverse variable instructions would add practical insights
>
> We appreciate this question and agree that scalability is key for real-world deployment. While our method requires precomputing steering vectors for specific instructions, this process is lightweight and requires only a small set of base queries (e.g., 20 for keyword-specific instructions), which can be unrelated to the inference-time question. This flexibility allows for efficient computation of steering vectors on the fly for novel instructions, making the approach practical for diverse user scenarios.
>
> Furthermore, real-world user instructions often combine multiple constraints (e.g., “The output should not exceed 5 sentences and must include a title”). Our results in Section 6 demonstrate that the steering procedure can handle such cases by breaking down the user request into distinct instructions and steering for each constraint simultaneously. This capability highlights the adaptability of our approach to complex, real-world tasks.
>
> ---
>
> In conclusion, we thank the reviewer for their constructive feedback, which has motivated valuable additions to our paper. With the new analyses, including the quality-accuracy trade-off and further discussions on scalability and cross-model transferability, we hope the reviewer will find the paper’s contributions even more compelling and will be open to increasing their overall rating.

---

> > ### Author Response · Authors · 2024-12-01
> >
> > As the discussion period is approaching its conclusion, we would like to thank the reviewer once again for their feedback, which has significantly contributed to improving our paper. During the rebuttal, we incorporated several of the reviewers’ suggestions, adding new results and analyses that addressed their points. This led to stronger results and 6 additional pages of content. We kindly ask if the reviewer is satisfied with our responses and whether they would consider revisiting their scores to reflect these improvements.

---

### Official Review · Reviewer_NNrP · 2024-11-09

**Soundness:** 3
**Presentation:** 3
**Contribution:** 3
**Rating:** 6
**Confidence:** 3

**Summary:**

This paper proposes a novel method to improve the instruction-following capabilities of language models using activation steering. Experiments on four different language models demonstrated the effectiveness of activation steering for three different tasks.

**Strengths:**

1. Clear presentation and writing.
2. The proposed method is simple and effective.
3. The experiments and analysis are comprehensive and solid.

**Weaknesses:**

1. The three instruction-following tasks are related to format, length, and word inclusion/exclusion, which are not broad enough for general instruction following cases.
2. No limitation or discussion section in the paper.

**Questions:**

1. line 159, "...perform a small grid search over neighboring values on a held-out set of examples to fine-tune the steering effect". How the examples are chosen? How many examples and what's the granularity of the grid? Does it need to be done for every layer?
2. Figure 4 is a bit confusing. Maybe it's better to mention how the delta is computed (e.g. after steering - original, w/ inst - w/o inst), instead of using "w/ vs. w/o", or simply "steering w/". Is this quality score a good metric? It's almost opposite with the results of accuracy.

---

> ### Author Response · Authors · 2024-11-21
>
> Thank you for acknowledging the effectiveness of our method and recognizing that our “experiments and analysis are comprehensive and solid.”
>
> > **Weakness 1**: The three instruction-following tasks are related to format, length, and word inclusion/exclusion, which are not broad enough for general instruction following cases.
>
> We would like to clarify that “format” is not a single instruction but a category that consists of 13 distinct instructions, each covering a specific output constraint (e.g., JSON formatting, punctuation rules, and casing). While the space of possible output constraints is indeed vast, we believe the tasks we consider are representative of many real-world use cases where specific formatting, brevity, or keyword constraints are critical.
>
> > **Weakness 2**: No limitation or discussion section in the paper.
>
> Although ICLR does not require a separate “Limitations” section, we acknowledge that a thorough discussion of limitations is good scientific practice. In the original submission, we addressed limitations in different parts of the paper, such as the discussion on steering side-effects in Appendix F.2. However, in response to this feedback, we have now added a dedicated “Limitations” section at the beginning of the Appendix.
>
> > **Q1**: How the examples are chosen? How many examples and what's the granularity of the grid? Does it need to be done for every layer?
>
> For keyword-specific instructions, we use GPT-4o to generate a set of 276 question-word pairs for validation. These examples are designed to resemble the base queries in IFEval, and are paired with relevant words that might appear in the responses. This synthetic dataset is then used for grid search during layer and weight selection.
> The grid of steering weights is centered around the average $c$ computed for each layer following Eq. 2. For example, for Phi-3 at layers 26 and 28, the average value of $c$ is approximately 42 and 52, respectively. We use this information to select the weights {40, 60, 80, 100} for the grid search. Layers {24, 26, 28} are evaluated systematically. The details of this procedure are now provided in Appendix E.
>
> > **Q2.1**: Maybe it's better to mention how the delta is computed (e.g. after steering - original, w/ inst - w/o inst)
>
> We appreciate the reviewer’s feedback. During this rebuttal phase, our focus has been on addressing the reviewers’ questions and performing the requested analyses. While we have not yet revisited the presentation of the paper, we will incorporate their suggestions in the next version.
>
> > **Q2.2**: Is this quality score a good metric? It's almost opposite with the results of accuracy.
>
> The quality score is specifically designed to measure the comprehensiveness of the response with respect to the base query (i.e., without considering the instructions). As some instructions inherently limit the comprehensiveness of the response (e.g., length constraints), it is expected that improvements in instruction-following accuracy may coincide with decreases in quality scores.
> This trade-off is evident in the rightmost bars of Figure 4, where explicit instructions (without steering) often lead to lower quality scores compared to outputs without instructions.
> As discussed in the general response, we have also updated our methodology to include a validation-time perplexity-based check, allowing us to improve the trade-off between accuracy and quality. This refinement is reflected in our updated results, where we demonstrate that steering can preserve most of the gains in instruction-following accuracy while minimizing quality degradation.
>
> ---
>
> We thank the reviewer for their thoughtful feedback and valuable suggestions. We hope that our responses have addressed their concerns and provided sufficient clarification.

---

> ### Author Response · Authors · 2024-12-01
>
> As the discussion period is approaching its conclusion, we would like to thank the reviewer once again for their feedback, which has significantly contributed to improving our paper. During the rebuttal, we incorporated several of the reviewers’ suggestions, adding new results and analyses that addressed their points. This led to stronger results and 6 additional pages of content. We kindly ask if the reviewer is satisfied with our responses and whether they would consider revisiting their scores to reflect these improvements.

---

### Author Response · Authors · 2024-11-21
**General Response**

We thank the reviewers for their positive and constructive feedback. In response to the reviewers’ suggestions, we conducted several additional experiments and analyses, which we believe significantly strengthen our paper. Below, we summarize the key modifications and additions we made. All changes in the revised manuscript are marked in blue for clarity.

**Mitigation of Quality Degradation through Perplexity-Based Checks**

A common point highlighted by multiple reviewers, particularly R96q, was the importance of preserving output quality during activation steering. Quality preservation is indeed a well-known challenge for steering methods. To address this, we improved our methodology by incorporating a general perplexity-based check into the validation process for layer and weight selection. This systematic approach is agnostic of the instruction type and enables us to maximize instruction-following accuracy while minimizing quality degradation. The revised results in the paper demonstrate an improved trade-off between accuracy and quality, with most of the accuracy gains preserved while mitigating drops in quality score. Details of this procedure are provided in Appendix E, and we updated Figures 3, 4, 6, 7, and 8 in the main paper, as well as Figure 9 in the appendix to reflect these improvements.

**Additional Analyses in Response to Reviewer XmmQ**

Reviewer XmmQ suggested additional analyses to provide deeper insights into our method. In response, we included the following in the revised paper:
- A detailed breakdown of instruction-following accuracy for specific format instructions across all models and settings (Appendix I).
- A breakdown of quality scores by instruction type to better illustrate the impact of steering (Appendix F.3).
- An analysis of the trade-off between output quality and instruction-following accuracy as mediated by the steering weight (Appendix F.4).
- Analyses of the geometry of the steering vectors, including t-SNE projections to show the separability of instruction representations (Appendix H).
- New experiments demonstrating the effectiveness of steering for exact-length constraints (e.g., “answer using $n$ sentences”) with statistically significant improvements (Appendix J).
- Evidence supporting the effectiveness of steering for word exclusion by subtracting word-inclusion vectors, showing it outperforms directly adding word-exclusion vectors (Appendix K).

---

### Meta-Review · Area_Chair_pAW2 · 2024-12-21

**Metareview:**

## Summary
This paper proposes a method to enhance instruction-following in language models using activation steering. The proposed method computes steering vectors by contrasting model activations from inputs with and without instructions. These vectors are used at inference time to adjust outputs according to specified constraints, such as format, length, or word inclusion. The method is evaluated on four models, showing its ability to improve constraint adherence, even without explicit instructions, and enhance performance when instructions are present. The study also explores combining multiple steering vectors and demonstrates that vectors derived from instruction-tuned models can be applied effectively to base models. This work provides a scalable technique for achieving fine-grained control in language generation.

## Decision

This paper studies a vital problem and addresses a challenging issue with LLMs. The findings and experiments in the paper, such as combining steering vectors and generalization to the base model, are intriguing. The paper is very well-written and has a clear presentation. The experiments and analysis are comprehensive and sound. Thus I recommend the paper for acceptance.

**Additional Comments On Reviewer Discussion:**

Overall, the reviewers unanimously agreed to accept this paper. The authors did a good job during the rebuttal, and as a result, some of the reviewers, like XmmQ, raised their scores. As a result, I recommend this paper for acceptance.

---

### Decision · Program_Chairs · 2025-01-22

Accept (Poster)